# Crystal structures of dimeric and heptameric mtHsp60 reveal the mechanism of chaperonin inactivation

Meng-Cheng Lai[1], Hao-Yu Cheng[1] , Sin-Hong Lew[1], Yu-An Chen[1], Chien-Hung Yu[2,5], Han-You Lin[3], Shih-Ming Lin[1,4]

Mitochondrial Hsp60 (mtHsp60) plays a crucial role in maintaining the proper folding of proteins in the mitochondria. mtHsp60 self-assembles into a ring-shaped heptamer, which can further form a double-ring tetradecamer in the presence of ATP and mtHsp10. However, mtHsp60 tends to dissociate in vitro, unlike its prokaryotic homologue, GroEL. The molecular structure of dissociated mtHsp60 and the mechanism behind its dissociation remain unclear. In this study, we demonstrated that *Epinephelus coioides* mtHsp60 (EcHsp60) can form a dimeric structure with inactive ATPase activity. The crystal structure of this dimer reveals symmetrical subunit interactions and a rearranged equatorial domain. The α4 helix of each subunit extends and interacts with its adjacent subunit, leading to the disruption of the ATP-binding pocket. Furthermore, an RLK motif in the apical domain contributes to stabilizing the dimeric complex. These structural and biochemical findings provide new insights into the conformational transitions and functional regulation of this ancient chaperonin.

## Introduction

Heat shock protein 60 (Hsp60), also known as chaperonin, is a widely conserved gene found in all living kingdoms (Zeilstra-Ryalls et al, 1991; Gupta, 1995; Bukau & Horwich, 1998; Zhao & Liu, 2017). Chaperonins play a crucial role in assisting protein folding and preventing protein aggregation in cells. There are two groups of chaperonins: type I, found in bacteria, mitochondria, and chloroplasts, and type II, found in eukaryotic cytosol and archaea (Llorca et al, 1999; Horwich et al, 2007; Vilasi et al, 2017). Mitochondrial Hsp60 (mtHsp60), a type I chaperonin, functions in the mitochondria with its co-chaperonin, mtHsp10 (Cheng et al, 1989; Ostermann et al, 1989). Several proteins critical for mitochondrial function are imported as unfolded single polypeptides and require folding with the assistance of mtHsp60 (Hartl, 1996; Frydman, 2001). One such protein is the F-type ATPase, which is crucial for ATP

synthesis (Prasad et al, 1990). In addition, mtHsp60 also helps mitochondrial proteins maintain proper folding without being denatured by the highly oxidative environment (Horwich et al, 1999; Voth & Jakob, 2017). As a result, mtHsp60 plays a vital role in maintaining mitochondrial functions in cells.

mtHsp60 is translated in cytosol as a precursor, called naïve mtHsp60, with a mitochondrial importing signal (MIS) at the N-terminus of 26 amino acids. Upon sorting to the mitochondria, the MIS sequence is removed, and naïve mtHsp60 matures into a 60 kD single polypeptide (Singh et al, 1990). mtHsp60 can self-assemble into a ring-shaped heptamer with seven identical protomers (Klebl et al, 2021; Wang & Chen, 2021). In the presence of mtHsp10 and ATP, these heptamers could stack back-to-back to form a double-ring tetradecamer (Nisemblat et al, 2015; Gomez-Llorente et al, 2020). Both single-ring and double-ring mtHsp60 complexes have ATP-dependent folding activities, suggesting that heptameric mtHsp60 is an active oligomeric state (Nielsen & Cowan, 1998; Gomez-Llorente et al, 2020).

The crystal structure of mtHsp60 reveals three domains: apical, intermediate, and equatorial domains (Nisemblat et al, 2015). The apical domain is a single segment that interacts with mtHsp10 and recruits unfolded polypeptides for the folding process. The intermediate domain connects the equatorial and apical domains and provides flexibility for mtHsp60's conformational changes. The equatorial domain connects the subunits in the heptameric ring and facilitates interactions between the two rings of tetradecamers. In addition, the equatorial domain contributes to the formation of the ATP-binding pocket, which involves the DGTTT motif ([85]DGTTT[89]) and residues G414 and D494 (Gomez-Llorente et al, 2020). These residues are known to participate in ATP binding in *Escherichia coli* GroEL and human mtHsp60 (Boisvert et al, 1996; Nisemblat et al, 2015; Gomez-Llorente et al, 2020). Upon ATP binding, mtHsp60 undergoes a conformational change that enables it to interact with mtHsp10 and assist in protein folding activity.

In addition to the heptameric and tetradecameric complex, various oligomeric states of mtHsp60 have also been reported, including octamer, dimer, and monomer (Klebl et al, 2021). Unlike other type I chaperonin homologs in bacterial, chloroplast, and

---

[1]Department of Biotechnology and Bioindustry Sciences, National Cheng Kung University, Tainan, Taiwan [2]Department of Biochemistry and Molecular Biology, National Cheng Kung University, Tainan, Taiwan [3]Department of Veterinary Medicine, School of Veterinary Medicine, National Taiwan University, Taipei, Taiwan [4]Institute of Tropical Plant Sciences and Microbiology, National Cheng Kung University, Tainan, Taiwan [5]Institute of Basic Sciences, College of Medicine, National Cheng Kung University, Tainan, Taiwan

Correspondence: smlin@mail.ncku.edu.tw

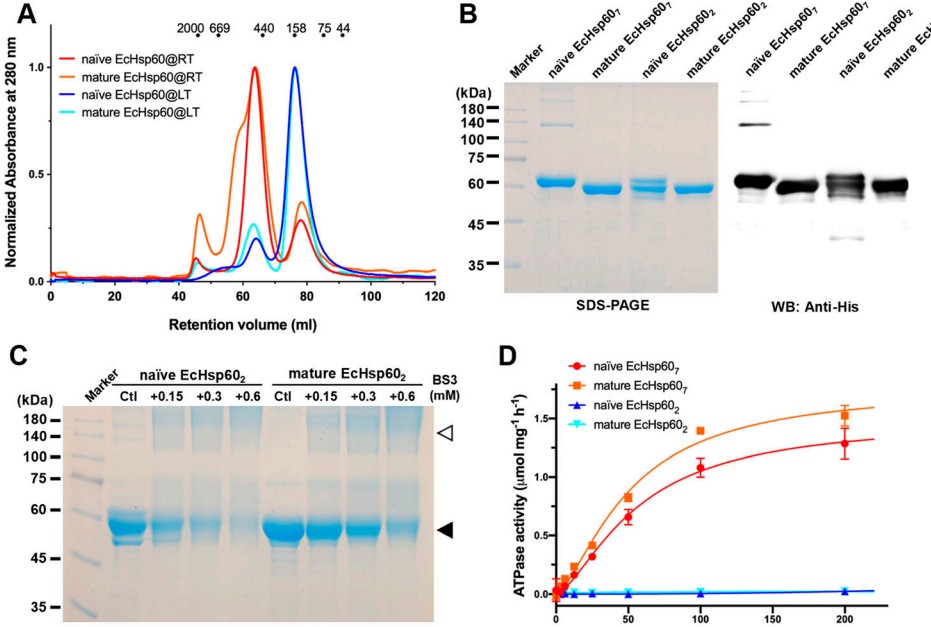

**Figure 1. Both naïve and mature EcHsp60 form dimers when purified at low temperature.**
**(A)** SEC analysis of the naïve and mature EcHsp60 purified under room temperature (RT, red and orange) and 4°C (blue and cyan), respectively. The elution volumes for different sizes of calibration markers are marked as dots on top of the chromatogram. **(B)** SDS–PAGE and Western blot analysis of the dimers (EcHsp60$_2$) and heptamers (EcHsp60$_7$) for naïve and mature EcHsp60. **(C)** Cross-linking analysis of dimeric EcHsp60. Both naïve and mature EcHsp60 in dimeric form were incubated with a series concentration of BS$^3$ and analyzed by SDS–PAGE. The filled triangle indicates the monomer of EcHsp60, whereas open triangle represents the dimeric EcHsp60. **(D)** Both EcHsp60$_2$ and EcHsp60$_7$ were used to measure the kinetic curves of ATPase activity at a range of ATP concentration from 3.06–200 $\mu$M. Each value represents the average of three independent measurements, and the error bars are standard deviations.

yeast mitochondria, mammalian mtHsp60 tends to dissociate into monomers during purification and is relatively unstable, whereas stable tetradecamers can be purified from other homologs (Viitanen et al, 1998; Levy-Rimler et al, 2001). A recent study suggests that mtHsp60 has shorter inter-subunit $\beta$-strands and unique residues at the inter-subunit interface, which decrease complex association and subunit interactions (Wang & Chen, 2021). Monomeric mtHsp60 is unable to hydrolyze ATP and loses its folding function (Viitanen et al, 1998). However, inactive monomeric mtHsp60 can reconstitute into oligomers by incubating with Mg-ATP. The biochemical characteristics and physiological functions of these dissociated mtHsp60 are not fully understood, and the molecular structure of mtHsp60 in its monomeric or dimeric form has also not been reported.

To address these gaps in knowledge, we cloned and expressed the mtHsp60 gene, *HSPD1*, from grouper fish (*Epinephelus coioides*) for biochemical and structural analysis. Our results showed that the *E. coioides* mtHsp60 (EcHsp60) forms dimers that are incapable of hydrolyzing ATP but can reassemble into heptamers, thus restoring ATPase activity. Furthermore, we reported the crystal structure of dimeric naïve EcHsp60, which reveals a disrupted ATP-binding pocket. Based on these results, we proposed a potential role for dimeric EcHsp60 in elucidating the conformational changes that occur in this vital chaperonin.

## Results

### Both the naïve and mature EcHsp60 formed a dimeric conformation after purification at low temperature

The stability and oligomeric state of mtHsp60 can be influenced by various factors, including protein concentration, the presence of nucleotides, and the existence of MIS (Levy-Rimler et al, 2001; Vilasi

et al, 2014; Ricci et al, 2016). In addition, previous studies have shown that heptameric mtHsp60 can significantly disassemble during purification at low temperatures (Viitanen et al, 1992, 1998). To study the oligomeric forms of EcHsp60, both the naïve and mature forms were expressed and purified at low temperature (LT, 4°C) and room temperature (RT, 25°C), respectively. Size-exclusion chromatography (SEC) was used to analyze the purified proteins, and it was found that both forms of EcHsp60 formed a major peak at a retention volume of 76 ml when purified at low temperature, indicating the formation of dimers (Fig 1A, blue and cyan lines). In contrast, when purified at room temperature, both proteins were predominantly eluted at 63 ml, indicating the formation of larger complexes, possibly heptamers (Fig 1A, red and orange lines). The presence of a shoulder peak only in the mature EcHsp60 purified at room temperature suggests the existence of larger oligomeric forms. These findings suggest that the oligomeric state of EcHsp60 can be affected by the purification temperature and that it can form dimers at low temperatures and heptamers at higher temperatures.

To confirm the protein composition of the major peaks observed in SEC, SDS–PAGE and Western blot analysis were performed (Fig 1B). The results revealed that all major peaks were composed of a single 60 kD polypeptide, indicating that the oligomeric states are composed solely of EcHsp60 without any host proteins. Negative-stain transmission electron microscopy (TEM) analysis further confirmed the heptameric structures of both naïve and mature EcHsp60, as observed in the ring-shaped structures consisting of seven subunits (Fig S1). In addition, cross-linking assays using BS$^3$ as the cross-linker confirmed the dimeric structures of both naïve and mature EcHsp60 (Fig 1C). The dimeric bands increased as the BS$^3$ concentration increased, providing further evidence of the formation of different oligomeric states of EcHsp60 depending on the purification temperature.

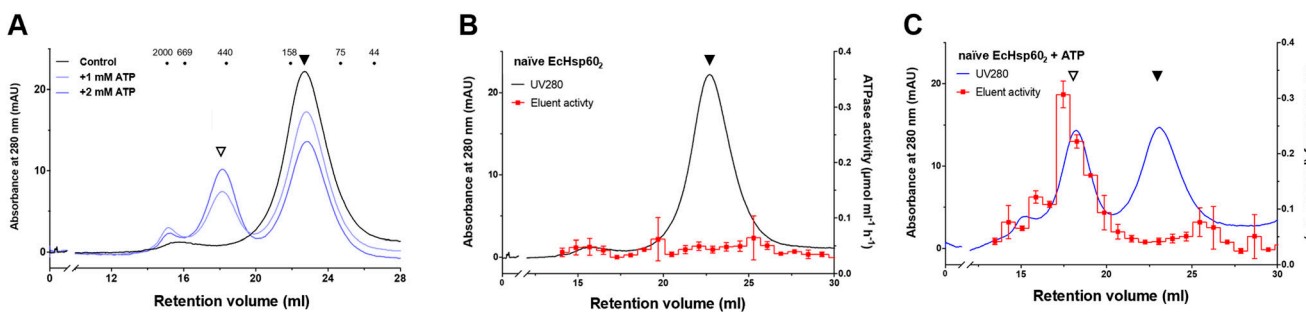

**Figure 2. Dimeric naïve EcHsp60 could be reconstituted into heptamers with restored ATPase activity.**
**(A)** SEC chromatogram showed the conformational changes of naïve EcHsp60$_2$ after incubation with various concentrations of ATP for 30 min at 30°C. **(B, C)** The naïve EcHsp60$_7$ assembled from the naïve EcHsp60$_2$ was separated and collected using SEC for measuring the ATPase activity. (B) is the naïve EcHsp60$_2$ only; (C) is the naïve EcHsp60$_2$ incubated with 4 mM ATP. The naïve EcHsp60$_2$ and assembled naïve EcHsp60$_7$ were indicated by filled and open triangles, respectively.

## Dimeric EcHsp60 is deficient in ATPase activity and shows lower stability

To further investigate the functional differences between the various oligomeric states of EcHsp60, the ATPase activity of EcHsp60$_7$ and EcHsp60$_2$ was measured. The kinetic profiles of both forms were determined over a range of ATP concentrations from 3.06–200 μM. It was found that EcHsp60$_7$ was capable of hydrolyzing ATP and its enzymatic kinetics were calculated by fitting a non-linear regression to allosteric sigmoidal equations (Fig 1D). The $K_{half}$ and $V_{max}$ values are reported in Table S1. The Hill slopes for both curves are 1.5, indicating a positive cooperative effect for EcHsp60$_7$. In contrast, neither the naïve nor the mature EcHsp60$_2$ showed any ATP hydrolysis activity at any ATP concentration (Fig 1D). These results indicate that EcHsp60 loses its ATPase function when forming a dimeric conformation.

In addition, it was observed that the naïve EcHsp60$_2$ degraded more quickly than other EcHsp60 species (Fig 1B), suggesting its lower stability. To assess the protein stability of naïve EcHsp60, a limited trypsin digestion assay was performed. The naïve EcHsp60$_7$ and EcHsp60$_2$ were incubated with trypsin, and the time-dependent degradation profiles were monitored using SDS–PAGE and Western blot analysis. The results showed that naïve EcHsp60$_7$ remained largely unchanged after 60 min of digestion, whereas the naïve EcHsp60$_2$ was significantly degraded to 50 kD within 15 min (Fig S2A). The Western blot analysis further confirmed that the C-terminal His-tag was still present, suggesting that the N-terminus of naïve EcHsp60$_2$ was degraded within 5 min of trypsinolysis (Fig S2B and C). These results indicate that the naïve EcHsp60$_2$ loses its ATPase activity and also reduces its protein stability.

## Naïve EcHsp60$_2$ could be reconstituted into heptameric complexes with restored ATPase activity

Previous studies have mentioned that the monomeric mtHsp60 can assemble into heptamers in the presence of ATP (Viitanen et al, 1998). To further understand this process, we investigated whether EcHsp60$_2$ could be reconstituted into an oligomeric state when ATP was added. The naïve EcHsp60$_2$ was incubated with 1 and 2 mM ATP for 30 min at 30°C, and its oligomeric state was monitored by SEC. The results showed that the naïve EcHsp60$_2$ reassembled into heptamers in the presence of ATP, with the proportion of heptamers increasing as the ATP concentration was raised (Fig 2A). In addition, the ATP hydrolysis activity was restored in the heptamers assembled from dimers, whereas the unassembled dimers remained inactive in ATP hydrolysis (Fig 2B and C). These findings suggest that the naïve EcHsp60$_2$ adopts a transient state with a loss of ATPase function, which can be reactivated into heptamers upon ATP binding.

## Crystal structure of the naïve EcHsp60$_2$ showed a twofold symmetric conformation with a reorganized equatorial domain

To understand the molecular basis of the inactivation of dimeric EcHsp60, the crystal structure of the naïve EcHsp60$_2$ was solved at a resolution of 2.35 Å (Table S2). This represents the first-ever structure of dimeric mtHsp60. The dimeric structure of the naïve EcHsp60 (EcHsp60$_2$) revealed a twofold symmetrical conformation, with two protomers facing each other in an asymmetric unit (Fig 3A). The N-terminal (MIS and residues 1–87) and the C-terminal (residues 508–552) regions of both protomers were not observed in the density maps because of their disordered structure, and only a partial equatorial domain was built in this model (Fig 3A). In contrast, the apical and intermediate domains of naïve EcHsp60$_2$ were well defined, except for a few flexible loops. The two protomers had a similar protein structure with a Cα root-mean-square deviation of 1.54 Å (Fig 3B). The primary structural variation between the two protomers is located at an α-helix section (T88-S106), which is also known as the fourth α-helix in the full-length chaperonin and will be referred to as α4 in the following discussion (Fig S3). The equatorial domain of the naïve EcHsp60$_2$ showed greater flexibility compared with the apical domain, as indicated by a higher average b-factor (Fig S4).

The analysis of the assembly interface revealed a favorable interaction between the two protomers of naïve EcHsp60$_2$, with a free energy of approximately –32.8 kcal/mol (Table S3). The interface between the two protomers, located at both the apical and equatorial domains, had a buried surface area of 3,943.8 Å$^2$ (Fig 3C and Table S3). Although hydrophobic contacts were dominant at the interface, several polar interactions were also present (Tables S4 and S5). The α4 helix of each subunit protruded out from its equatorial domain to interact with the adjacent subunit, with the

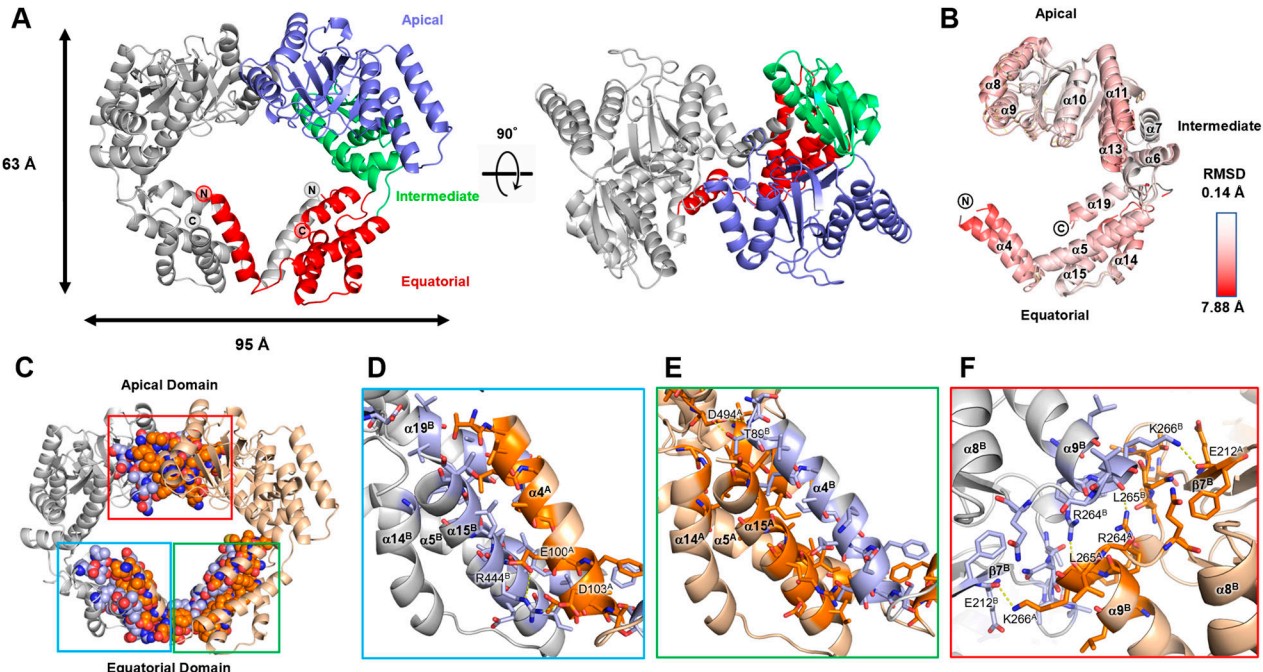

**Figure 3. Crystal structure of the naïve EcHsp60₂ revealed a face-to-face symmetric conformation.**
**(A)** Side view and top view of the overall structure of naïve EcHsp60₂. Chain A is colored by the domain and chain B is colored in silver. N- and C-termini of both subunits are labeled. **(B)** Superposition of the two protomers of the naïve EcHsp60₂. Coordinates are shown in cartoon and colored by RMSD values which are calculated by Pymol. **(C)** The interface residues between two protomers are shown in sphere and colored in deep orange and light blue for chains A and B, respectively. The atom types are colored in CPK to reveal the interacting forces. **(D, E, F)** The zoom-in views of the blue, green, and red boxes in (C), respectively. (D, E) focus on the interface between the α4 helices and their adjacent subunit, whereas (F) shows the top view of the interface at the apical domain. The residues involved in the interface are shown in sticks. The polar interactions are shown in yellow dash lines, and the residues involved in polar interaction are labeled. The chain ID of each residue and the α-helices are labeled in superscript.

interface mainly composed of van der Waals forces and hydrophobic interactions (Fig 3D and E and Table S5). Two intermolecular polar contacts were identified on the α4 helix of each subunit, involving residues T89, E100, D103, R444, and D494 (Fig 3D and E and Table S4). These residues are conserved in various types of chaperonins, except for D103 (Fig S3). Interestingly, the inter-subunit interactions on the α4 helix differed between the two subunits, suggesting that α4 may have flexible contacts with the adjacent subunit in various binding modes. In contrast, the apical domain formed symmetrical contacts between the two protomers (Fig 3F and Table S4). The RLK motif (R264-K266) located at the α9 terminus formed four hydrogen bonds with residues from another protomer, including E212, R264, L265, and K266 (Fig 3F and Table S4). The RLK motif is conserved in animal mtHsp60, as revealed by sequence alignment analysis (Fig S3). These interacting residues contribute to the formation of the dimeric complex of EcHsp60.

Notably, a strong extra-density was identified near the dimer interface of naïve EcHsp60₂ in the Fo-Fc maps (higher than 10 σ) (Fig S5). This extra-density is surrounded by the side chains of two Y201 residues from both protomers and is near the RLK motif. Although the exact identity of the molecule(s) in this extra-density cannot be determined, it is suggested that they have a high occupancy and may play a role in the formation of dimeric EcHsp60. This crystal structure provides novel insight into the mechanism of dimerization of mtHsp60.

## Structural comparison between the dimeric and heptameric EcHsp60 reveals the impact of conformational changes on the ATP-binding pocket

To better understand the conformational changes between the dimeric and heptameric EcHsp60, we attempted to crystallize the naïve EcHsp60₇. Unfortunately, we obtained poorly diffracting crystals of naïve EcHsp60₇. As an alternative, we truncated the highly dynamic C-terminal GGM repeats of naïve EcHsp60 (Δ527–552), based on the crystallographic study of human mtHsp60 (Nisemblat et al, 2015). The resulting truncated protein, called naïve EcHsp60$^{\Delta C}$, was purified as a stable heptameric complex at both 4°C and RT (Fig S6A). Importantly, the heptameric naïve EcHsp60$^{\Delta C}$ showed similar ATPase activity to the WT naïve EcHsp60₇ (Fig S6B), indicating that the C-terminal truncation did not significantly affect the protein structure. Therefore, we used these truncated naïve EcHsp60$^{\Delta C}$ as the heptameric EcHsp60₇ for subsequent structural studies.

These truncated naïve EcHsp60₇ proteins were successfully crystallized, and the diffraction power of the crystals dramatically improved to a resolution of 3.5 Å (Table S2). The crystal structure of naïve EcHsp60₇ showed a typical heptameric, single-ring conformation in an asymmetric unit (Fig 4A). Unlike the partially disordered equatorial domain of naïve EcHsp60₂, most regions of naïve EcHsp60₇ could be observed in the electron density maps, except for the N-terminal MIS and two C-terminal residues (residues

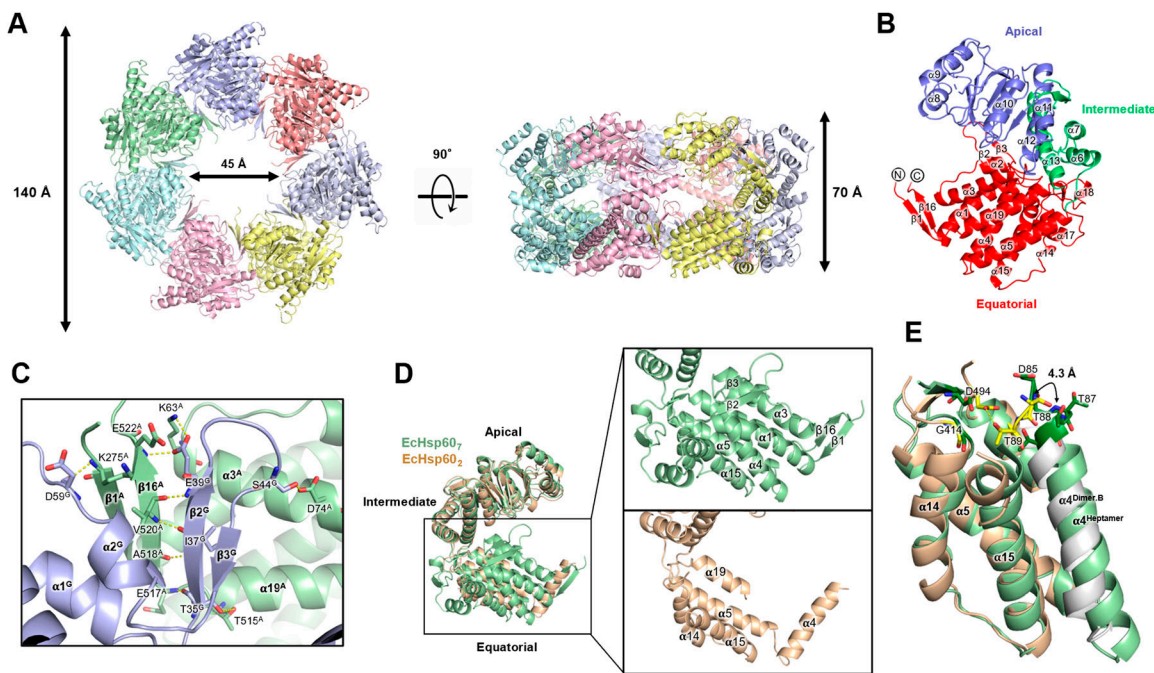

**Figure 4. Crystal structure of the naïve EcHsp60₇ reveals a significant conformational difference at the equatorial domain compared with the naïve EcHsp60₂.**
**(A)** The top and side views of naïve EcHsp60₇ structure. The coordinates are shown in cartoon and colored by chain. The molecular sizes are labeled. **(B)** The protomer is colored by the domain. N- and C-termini of both subunits are labeled. **(C)** EcHsp60₇ formed two anti-parallel β-strand pairs to connect the adjacent protomers. The residues involved in inter-molecular interaction are shown in sticks and labeled. **(D)** Superimpose of the protomers of naïve EcHsp60₂ (wheat) and naïve EcHsp60₇ (green). Significant conformational changes at the equatorial domain are zoomed in the right panel. The secondary structures were labeled to indicate the differences. **(E)** Structural comparison of the equatorial domains of the naïve EcHsp60₂ (wheat for chain A and silver for chain B) and naïve EcHsp60₇ (green). The residues involved in binding ATP are shown in sticks and colored in deep green (for naïve EcHsp60₇) and yellow (for naïve EcHsp60₂).

525–526) (Fig 4B). The equatorial domain consists of 11 α-helices (α1-α5, α14-α19) and three sets of anti-parallel β-strands (β1/β16, β2/β3, β14/β15). The β1/β16 strands form a strong interaction with the β2/β3 strands of the neighboring protomer and stabilize the heptameric complex (Fig 4C). The thermal parameter distribution of the naïve EcHsp60₇ coordinate showed relatively low b-factor values in the equatorial domain compared with the apical and intermediate domains, suggesting that the equatorial domain is stabilized upon forming the heptameric conformation (Fig S4).

The structural comparison between the heptameric and dimeric EcHsp60 reveals differences in the equatorial domains, whereas the apical and intermediate domains showed similar conformations in both oligomeric states (Fig 4D). The α4 helix of naïve EcHsp60₇ was found to be folded into the core region of the equatorial domain with α1, α5, α16, and α20, whereas in naïve EcHsp60₂, the α4 helix extended out from the equatorial domain and interacts with the neighboring subunit (Fig 4D). In addition, the α1-3, α17-α19, and all β strands in the equatorial domain were only observed in naïve EcHsp60₇, suggesting that the equatorial domains may dynamically fold during the formation of the heptameric ring structure (Fig 4D).

To better understand the loss of ATPase activity in EcHsp60₂, the ATP binding sites were evaluated in the structures of naïve EcHsp60₂ and EcHsp60₇. In naïve EcHsp60₇, the ATP-binding motif, DGTTT, is located in the N-terminus of α4 helix and forms a pocket with G414 and D494 (Fig 4E). However, in EcHsp60₂, the α4 helix from the neighboring subunit is slightly twisted, resulting in the

rearrangement of the ATP-binding residues. The T89 forms hydrogen bonds with D494, occupying the space of the ATP-binding pocket, and the D85, G86, and T87 are disordered, leading to the inability of EcHsp60₂ to hydrolyze ATP. These structural differences between naïve EcHsp60₂ and EcHsp60₇ provide an explanation for the inactivation mechanism of mtHsp60. The conformational dynamics of the equatorial domain may play a crucial role in the functional regulation of mtHsp60.

Moreover, we compared the structure of naïve EcHsp60₇ with well-studied chaperonins, including human mtHsp60 and *E. coli* GroEL (Fig S7). These proteins exhibit a similar ring-shaped arrangement of seven protomers (Fig S7A and D) and a comparable overall conformation, with slight variations in the apical domain (Fig S7B and E). The α4 helix is well folded in the equatorial domain of all heptameric chaperonins, and the β1/β16 and β2/β3 responsible for subunit interaction also have a similar conformation between these chaperonin heptamers (Fig S7C and F). These structural similarities support the idea that naïve EcHsp60, containing MIS, can still oligomerize into heptamers with a conserved protein conformation.

## Decreased stability of dimeric interaction promotes the formation of heptamers in EcHsp60

The crystal structure of naïve EcHsp60₂ showed that both the α4 helix and the RLK motif contribute to forming inter-subunit interactions (Fig 3F). To further investigate the role of the RLK motif in

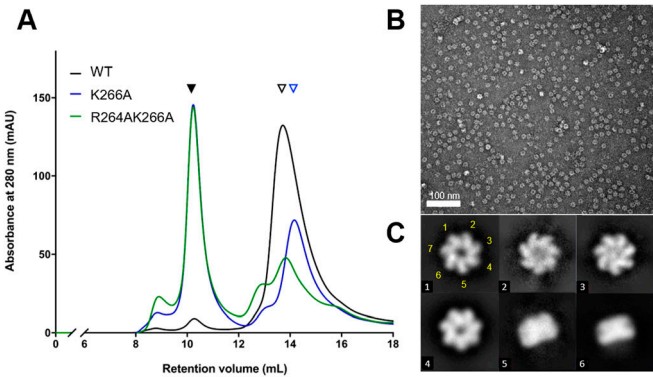

**Figure 5. The RLK motif is essential for maintaining the dimeric conformation of naïve EcHsp60.**
**(A)** The WT and mutated naïve EcHsp60 were purified at 4°C for the SEC analysis. Black filled and open triangles represent the heptameric and the dimeric EcHsp60, respectively. The blue open triangle indicates the monomeric EcHsp60. **(B)** Negative-stain TEM micrograph of the R264AK266A heptamers in the SEC eluent. **(C)** Representative 2D classification results showed the single-ring heptameric conformation of R264AK266A. The class number of each particle is marked in the lower left corner. The classes #5 and #6 show the side view of the single-ringed heptamer. The position of the seven subunits in the class #1 particle is marked with yellow numbers.

the formation of dimeric EcHsp60, we generated point mutants (R264A and K266A) and a double mutant (R264AK266A) of the RLK motif. These mutants were expressed in *E. coli* and purified under both 4°C and RT conditions. However, the R264A mutant showed significant precipitation during purification, indicating that its solubility or stability may be affected. Therefore, only the K266A and R264AK266A mutants were further analyzed by SEC and compared with the WT naïve EcHsp60 (Fig 5A). The SEC results showed that the EcHsp60$_7$ peak was significantly increased in the K266A and R264AK266A mutants compared with the WT, whereas the peak of naïve EcHsp60$_2$ was dramatically decreased in the K266A and R264AK266A mutants. Furthermore, the retention volume of the dissociated EcHsp60 was slightly increased in the K266A and R264AK266A mutants, suggesting that these mutants may not form dimers but rather monomers (Fig 5A, blue triangle). The oligomeric conformation of the R264AK266A mutant was confirmed by negative-stain TEM, which showed that it formed a heptameric ring structure similar to the WT naïve EcHsp60$_7$ (Fig 5B and C). These results suggest that the RLK motif is crucial for stabilizing the dimeric form of EcHsp60.

## Discussion

This study represents a significant advancement in our understanding of a novel conformation of chaperonin. The findings indicate that EcHsp60 can form a dimeric conformation in which the equatorial domain is partially reorganized, and the ATP binding pocket is disrupted, leading to an inactive state. The crystal structure showed that the RLK motif and the α4 helix play a crucial role in stabilizing the EcHsp60 dimer through intermolecular interactions. Upon incubation with ATP, the inactive dimer of EcHsp60 can be reconstituted into a heptameric ring complex, restoring its

ATPase activity. This suggests that the dimeric complex is a temporary inactive state for EcHsp60.

Based on these findings, we have proposed a working model to elucidate the conformational transition during EcHsp60 oligomerization from dimer to heptamer (Fig 6). The α4 helix of each monomer is exchanged to link two protomers together in a face-to-face orientation to form a dimer (Fig 6A). The large conformational change at α4 helix disrupts the ATP-binding pocket and renders the dimeric EcHsp60 unable to bind ATP. The dimers might dynamically dissociate into individual monomers before forming the heptamers in a circular arrangement (Fig 6B). Upon ATP binding, the monomeric EcHsp60 forms a proper ATP-binding pocket with the refolded α4 helix in the equatorial domain, leading to the formation of the heptameric ring-shaped complex (Fig 6C). Our proposed model provides insight into the molecular mechanisms of the ATP-stimulated assembly and the complex instability of mtHsp60.

In a previous study, it was shown that human naïve mtHsp60 could form stable heptamers in solution (Vilasi et al, 2014). Similarly, our study found that naïve EcHsp60 can form stable heptamers with comparable ATPase activity to mature EcHsp60. Structural analysis also revealed that the naïve EcHsp60$_7$ structure is similar to that of mature human mtHsp60. However, the SEC analysis showed that mature EcHsp60 forms a complex slightly larger than the heptameric complex but much smaller than 669 kD when purified at room temperature (Fig 1A). We proposed that this additional complex may be an octameric complex formed by a single-ring heptamer with a monomer in the central cavity (Klebl et al, 2021). The absence of octameric complexes in naïve EcHsp60 suggested that the presence of MIS may restrict substrate binding, despite not affecting the heptameric structure and ATPase activity. Although the MIS peptide was not observed in the naïve EcHsp60$_7$ because of its intrinsically disordered structure, it is expected to extend inside the heptameric ring and occupy the central cavity (Spinello et al, 2015). This cavity serves as a folding chamber for accommodating unfolded proteins during the folding process (Hartl, 1996). Therefore, the presence of MIS may limit the available space within the folding chamber of mtHsp60, potentially affecting its ability to bind to unfolded substrates. Further research is needed to examine the folding ability of naïve mtHsp60 to understand its role outside of the mitochondria.

Previous studies have reported the dissociation of both mitochondria and chloroplast Hsp60s into monomers at low protein concentrations, but there is limited structural information on monomeric chaperonins (Viitanen et al, 1998; Dickson et al, 2000; Parnas et al, 2009). In contrast, the prokaryotic chaperonin *Mycobacterium tuberculosis* Cpn60.2 (MtCpn60.2) has been observed to assemble into dimers in both solution and protein crystals, providing insights into the structure of dissociated chaperonins (Qamra & Mande, 2004; Shahar et al, 2011). Structural comparison between EcHsp60 and MtCpn60.2 revealed different arrangements of subunits in forming the dimeric complex (Fig S8). Specifically, the subunits in the naïve EcHsp60$_2$ dimer are symmetrically arranged, whereas the subunits of MtCpn60.2 interact through contacts between their apical and equatorial domains (Fig S8A–C). Compared with MtCpn60.2, naïve EcHsp60$_2$ forms a larger interface between subunits with more inter-subunit interactions, suggesting that EcHsp60$_2$ forms a more stable dimeric conformation. Nevertheless,

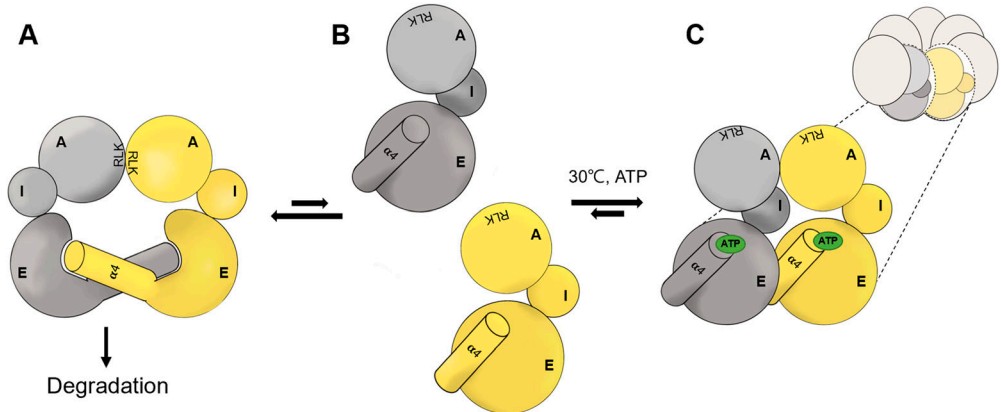

**Figure 6. Proposed mechanism for the reassembly of EcHsp60 from an inactive dimer to active heptameric ring. (A)** EcHsp60 formed a homodimer through the interaction of RLK motifs and the protruded α4 helix. The EcHsp60$_2$ is sensitive to degradation because of the partial unfolded equatorial domain. **(B)** The dynamically dissociated EcHsp60 forms a well-folded equatorial domain with an ATP-binding pocket at the terminus of α4 helix. **(C)** The EcHsp60 monomers assemble into single-ring heptamers with the stable equatorial domain bound to ATP. The ATP binding (green circle) would stabilize the heptameric conformation and push the dynamical equilibrium to the right. A, I, and equatorial domains, respectively. The gray and yellow colors indicate different protomers of EcHsp60.

in both EcHsp60$_2$ and MtCpn60.2, the N-terminal β1-3 and C-terminal β16 regions are not observed in the crystal structures (Fig S8D). These β-strands are responsible for connecting the subunits in the heptameric EcHsp60$_7$, and current structural evidence suggests that they may only form secondary structures upon oligomerization into the heptameric ring complex.

Furthermore, this study highlights a novel function of the apical domain in the chaperonin structure. In *E. coli* GroEL, the α8 and α9 helices are responsible for substrate recognition and binding by interacting with hydrophobic patches of the substrate peptides (Buckle et al, 1997; Chen & Sigler, 1999). In contrast, in dimeric EcHsp60, these helices contribute to inter-subunit interactions on the apical domain (Fig 3). In addition, the α9 helix also participates in the dimer formation of MtCpn60.2 (Fig S8E). Residue R266 in MtCpn60.2, corresponding to K266 in EcHsp60, forms hydrogen bonds with another subunit in the dimeric complex. These findings suggest that the apical domain could interact with various peptides to perform diverse functions, emphasizing the significance of the apical domain in chaperonin functions.

Overall, this study provides useful insights into the structural and functional aspects of mtHsp60. The identification of a novel dimeric conformation improves our knowledge on the molecular mechanisms of mtHsp60 assembly and reactivation. These findings contribute to our understanding of the chaperonin-mediated protein folding regulation. Further research is needed to fully elucidate the physiological functions of mtHsp60 in cells and to explore its potential as a therapeutic target.

## Materials and Methods

### Plasmids

The cDNA library of grouper fishes was prepared according to previous studies (Hsu et al, 2013). The -cDNA encoding EcHSP60 was amplified and cloned into the pET24a (Invitrogen) vector between the restriction enzyme cutting sites, NdeI and XhoI. The constructed plasmid was named pET-EcHSP60 and was used to express the recombinant EcHSP60 fused with a C-terminal six-His

tag. To express the C-terminal truncated EcHSP60 removing the flexible GGM tail, the DNA region encoding E527 to G552 was removed from pET-EcHSP60 to form pET-EcHsp60$^{\Delta C}$ using In-Fusion seamless cloning kit (Takara Bio). In addition, the plasmids used to express the R264A, K266A, and R264AK266A mutants were also constructed by using the In-Fusion seamless cloning kit based on pET-EcHSP60 as the template.

### Protein expression and purification

The constructed plasmids, pET-EcHsp60 and pET-EcHsp60$^{\Delta C}$, were respectively transformed into *E. coli* Rosetta (DE3) strain for large-scale protein production. The protein expression was induced by adding 0.5 mM IPTG into the *E. coli* culture during the mid-log phase at 30°C for 6 h. After EcHsp60 expression, the *E. coli* culture was harvested and resuspended in the lysis buffer (25 mM Tris–HCl, pH 7.6, 300 mM NaCl, 0.1 mM PMSF). Ultrasonication was employed for cell homogenization with a power intensity of 960 J/ml and a duty cycle of 1 s (UP200S; Hielscher). The EcHSP60$_2$ was homogenized and purified at 4°C, whereas EcHSP60$_7$ was carried out at room temperature. The C-terminal His-tagged EcHsp60 was purified using Ni-NTA columns on an FPLC system (ÄKTA pure; Cytiva). The purified proteins were further concentrated for SEC purification using the SEC buffer (25 mM Tris–HCl, pH 7.6 and 150 mM NaCl) at a flow rate of 1 ml/min. The size-exclusion column (Superdex S200pg, 16/600; Cytiva) was used for the large-scale preparation of heptameric and dimeric EcHSP60. For analytical preparation of mutant EcHsp60s, the high-resolution SEC column (Superdex 200 Increase 10/300 GL; Cytiva) was used instead. Purified EcHSP60 was concentrated to a final concentration of 10 mg/ml using a centrifuge concentrator (Amicon Ultra-15, 30 kD MWCO; Millipore). The concentrated EcHSP60$_2$ and EcHSP60$_7$ were stored at 4°C and room temperature, respectively, for following studies.

### Cross-linking assay

The cross-linking assay was performed with slight modifications to previously published protocols (Parnas et al, 2009; Shi et al, 2017). The dimeric EcHsp60 was dialyzed into the assay medium (4.56 mM

NaH$_2$PO$_4$, 20.44 mM Na$_2$HPO$_4$, and 300 mM NaCl, pH 7.6) and concentrated to 0.6 mg/ml for addition of the bis(sulfosuccinimidyl) suberate (BS$^3$; Thermo Fisher Scientific). For each reaction, 30 $\mu$l of protein and 10 $\mu$l of BS$^3$ were mixed to give a final concentration of BS$^3$ from 0.15 to 0.6 mM and incubated at 37°C for 30 min. After incubation, the cross-link reaction was quenched by adding 10 $\mu$l stop solution (1 M Tris–HCl, pH 8.0), and the samples were analyzed by 10% SDS–PAGE.

## ATP hydrolysis assay

The ATP hydrolysis assay is according to previous studies with minor modifications (Lin et al, 2012). Twenty micrograms of EcHSP60 were mixed with an assay medium (50 mM Tris–HCl, pH 7.5, 20 mM KCl, and 1 mM MgCl$_2$) containing various concentrations of ATP in a final volume of 90 $\mu$l and incubated at 37°C for 30 min. The reactions were stopped by adding 180 $\mu$l stop solution (0.7% [wt/vol] ammonium molybdate, 0.02% [wt/vol] 1-amino-2-naphthol4-sulfonic acid, 2.0% [wt/vol] sodium dodecyl sulfate, and 1.16 N HCl). The amount of P$_i$ released in each sample was quantified by measuring the absorbance at 700 nm using a microplate reader (Multiskan SkyHigh; Thermo Fisher Scientific). Twofold serial dilutions of P$_i$ standards were used to generate the calibration curve in the concentration range of 4–128 $\mu$M. Each sample was repeated three times to calculate the mean and SD. The kinetic curves were obtained by fitting the experimental data with a nonlinear regression equation of the substrate inhibition function using the software, GraphPad Prism 8.0. The K$_{half}$ and V$_{max}$ values with standard error were calculated from the fitting equations.

## Limited trypsin digestion

Aliquots of EcHSP60$_2$ and EcHSP60$_7$ were diluted in the protein buffer (25 mM Tris, 150 mM NaCl, pH 7.6) containing 50 units of trypsin to a final protein concentration of 1 mg/ml and incubated at 25°C for various time courses. The reactions were stopped by mixing with SDS–PAGE loading buffer to denature the trypsin and EcHsp60. All samples were analyzed by 10% SDS–PAGE and Western blot to detect the residual amount of EcHsp60. The primary and secondary antibodies are anti-His mouse antibody and anti-mouse IgG goat antibody, respectively.

## EcHSP60 assembly assay

The heptamer assembly method used for EcHsp60 is based on the reconstitution assay of oligomeric human mtHsp60 with some minor modifications (Viitanen et al, 1998). The purified EcHsp60$_2$ was diluted to 1.2 mg/ml in the assay medium (50 mM Tris, NaCl, pH 7.5, 300 mM NaCl, 20 mM Mg(CH$_3$COO)$_2$, and 20 mM KCl) containing various concentrations of ATP to trigger the assembly process. The samples were incubated at 30°C for 2 h, and then each sample was injected with 500 $\mu$l for SEC analysis using an FPLC system (ÄKTA pure; Cytiva) with the SEC column (Superdex 200pg 10/600; Cytiva). The SEC buffer (25 mM Tris–HCl, pH 7.6, and 150 mM NaCl) was used at a flow rate of 0.5 ml/min. UV absorbance was monitored at 280 nm. The eluents containing EcHSP60 were fraction-collected for further measurements of the ATPase activity.

## Protein crystallization

To crystallize EcHsp60, the EcHsp60$_2$ and EcHsp60$_7$ proteins were purified and concentrated to 50 mg/ml and 40 mg/ml in the SEC buffer (25 mM Tris–HCl, pH 7.6, and 150 mM NaCl), respectively. Initial crystallization conditions were identified from robotic screening using the hanging-drop vapor diffusion method at 20°C. Protein crystals were obtained from Wizard classic 2 NO. 48 (Rigaku) for EcHsp60$_2$ and from MemGold2 G2 (Molecular dimension) for EcHsp60$_7$. The optimized crystal condition for EcHSP60$_2$ is 400 mM potassium sodium tartrate and 21% (wt/vol) PEG 3350, whereas for heptamer, it is 50 mM Tris–HCl, 500 mM MgCl$_2$, 10.5% (wt/vol) PEG1000. The crystals of both proteins were grown in 7 d at 20°C.

## X-ray diffraction and data refinement

Diffraction data of the EcHsp60$_2$ and EcHsp60$_7$ crystals were collected at beamline BL13B1 of NSRRC. Each diffraction image was exposed for 20 s under a low-temperature nitrogen stream with a 1° oscillation for a total of 180 images. The diffraction data were indexed, integrated, and scaled using HKL2000 (Otwinowski & Minor, 1997). Molecular Replacement (MR) was conducted by using Phaser implemented in the Phenix to obtain the correct phase (Liebschner et al, 2019). The search model was the predicted model generated by SWISS-MODEL using human mtHsp60 (PDB:6MRD) as a template (Gomez-Llorente et al, 2020). The models obtained from phasing solutions were further refined and validated using Phenix and Coot. Ridge body refinement was first introduced to refine the initial models derived from MR. The "secondary structure restrains" and "Ramachandra restrains" are applied to maintain the geometry for heptameric EcHsp60. TLS restrain refinement was then applied to reduce the R$_{free}$ for both sets of diffraction data. The models after refinements are visualized using Pymol and Chimera (Pettersen et al, 2004). The inter-molecular polar interactions of the refined models were identified by PDBePISA analysis (Krissinel & Henrick, 2007).

## Negative-stain TEM and single particle analysis

The purified oligomeric EcHSP60 was diluted to 1 mg/ml, and 4 $\mu$l samples were placed on the glow discharged grids (Formvar/carbon-coated Cu grids, 400 mesh) for 60 s and then removed by filter paper blotting. The grids were washed by 8 $\mu$l ddH$_2$O drops for two times, and 2% (wt/vol) uranyl acetate was added to stain the grids for 60 s. After removal of the staining solution, the grid was air-dried completely for TEM analysis. The micrographs were acquired using a TEM (JEOL JEM-1400) at an electron voltage of 100 kV. Micrographs processed with CTF Estimation were used for automatic particle picking using cisTEM (Grant et al, 2018). The picked particles were then used for 2D classification and to build the initial map using cryoSPARC (Punjani et al, 2017). The initial map was further refined with C7 symmetry using homogeneous refinement. The density map fitted with mtHSP60 coordinates was displayed by using Chimera (Pettersen et al, 2004). For WT EcHsp60 heptamers, 24,177 particles were picked from 61 micrographs, and 15,685 particles selected after two runs of 2D classification were used to build

the initial map. For R264AK266A heptamers, 18,626 particles were picked from 40 micrographs, and 12,051 particles remained after the 2D classifications.

## Data Availability

The structural coordinates and structure factors of the dimeric and heptameric EcHsp60 have been deposited in the PDB (http://www.rcsb.org/) with the following accession numbers: 7V9R—heptameric naïve EcHsp60; 7V98—dimeric naïve EcHsp60.

## Supplementary Information

## Acknowledgements

We appreciate the experimental facility and the technical services provided by the Synchrotron Radiation Protein Crystallography Core Facility (SPXF) of the National Core Facility for Biopharmaceuticals (NCFB), Ministry of Science and Technology (MOST), and the National Synchrotron Radiation Research Center (NSRRC), a national user facility funded by the Ministry of Science and Technology (MOST), Taiwan (R.O.C.). Initial crystallization screening was carried out at the NSRRC-NCKU Protein Crystallography Laboratory at University Center for Bioscience and Biotechnology of National Cheng Kung University (funding by NSRRC 11123LAB02). We gratefully acknowledge the use of TEM instruments (EM000900 of MOST 110-2731-M-006-001) belonging to the Core Facility Center of National Cheng Kung University. We would like to acknowledge funding from the Ministry of Science and Technology (MOST) of Taiwan (MOST 109-2636-B-006 -012; MOST 110-2636-B-006 -012; MOST 111-2636-B-006 -012) to S-M Lin.

### Author Contributions

M-C Lai: data curation, formal analysis, and writing—original draft.
H-Y Cheng: data curation and formal analysis.
S-H Lew: data curation and formal analysis.
Y-A Chen: data curation, software, and formal analysis.
C-H Yu: conceptualization, resources, and writing—review and editing.
H-Y Lin: conceptualization, methodology, and writing—review and editing.
S-M Lin: conceptualization, supervision, and writing—original draft, review, and editing.

### Conflict of Interest Statement

The authors declare that they have no conflict of interest that could have influenced the results or interpretation of the findings presented in this study. All authors have participated in the design, execution, and analysis of the research and have approved the final manuscript for submission. No external funding or support was received from organizations that could have influenced the outcome of the study in any way.

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
