## [Reviewer comments · Life Science Alliance]

Life Science Alliance

Crystal structures of dimeric and heptameric mtHsp60 reveal the mechanism of chaperonin inactivation

Meng-Cheng Lai, Hao-Yu Cheng, Sin-Hong Lew, Yu-An Chen, Chien-Hung Yu, Han-You Lin, and Shih-Ming Lin
DOI: <https://doi.org/10.26508/lsa.202201753>

Corresponding author(s): Shih-Ming Lin, National Cheng Kung University

Review Timeline:

Submission Date:	2022-10-05
Editorial Decision:	2022-11-14
Revision Received:	2023-02-14
Editorial Decision:	2023-03-07
Revision Received:	2023-03-13
Editorial Decision:	2023-03-14
Revision Received:	2023-03-16
Accepted:	2023-03-16

Transaction Report:

November 14, 2022

Re: Life Science Alliance manuscript #LSA-2022-01753-T

Dr. Shih-Ming Lin
National Cheng Kung University
Department of Biotechnology and Bioindustry Sciences
1, University Rd,
East Dist.,
Tainan city, Tainan 701
Taiwan

Dear Dr. Lin,

Thank you for submitting your manuscript entitled "Crystal structures of dimeric and heptameric mtHsp60 reveal the mechanism of chaperonin inactivation" to Life Science Alliance. The manuscript was assessed by expert reviewers, whose comments are appended to this letter. We invite you to submit a revised manuscript addressing the Reviewer comments.

Thank you for this interesting contribution to Life Science Alliance. We are looking forward to receiving your revised manuscript.

Sincerely,

B. MANUSCRIPT ORGANIZATION AND FORMATTING:

Reviewer #1 (Comments to the Authors (Required)):

This study report the cloning, purification and structural analysis of recombinant mitochondrial Hsp60 from *Epinephelus coioides*. It is demonstrated that under low temperature the recombinant protein is isolated as low molecular weight component while at RT it is purified as heptamer. The system is under equilibrium, as in the presence of ATP it assemble into heptamers. The crystal structure of the dimeric form and the c-terminal truncated form is solved. The report of dimers and heptamers is of high interest, however, some issues need to be addressed before publications

- 1) Visually the crosslinking results show dimers and monomers - not homogenic dimers. This need to be carefully examined using AUC.
- 2) Was the histag removed at the end?. Do you expect it to interfere with assembly (at least from the structure of heptamer..
- 3) more discussion on differences between subunit contacts in various hsp60 and EC Hsp60.
- 4) If I remember well, structure of dimeric cpn60 from mycobacterium was published in the 90s. Can you see differences?
- 5) The term functional is misleading here. It is functional in ATP hydrolysis not in protein folding (this was not examined). Is it active in protein folding? Do you have in hand purified Hsp10 from EC? If not is it active with Hsp10 from other animals?

Reviewer #2 (Comments to the Authors (Required)):

The authors (Lai et al.) report two crystal structures of mitochondrial chaperonin mtHsp60 from *Epinephelus coioides* in dimeric and heptameric conformations. mtHsp60 is known to exist as an unstable heptamer which dissociates irreversibly to stable dimers. The structure of the dimer illustrates for the first time how the dimer is assembled and explains why the dimer does not hydrolyze ATP as the heptamer. The work provides important insights into the puzzling heptamer-to-dimer dissociation. However, the manuscript has numerous incorrect and even faulty statements, and needs major revisions. The following concerns need to be addressed.

Major concerns:

1. The protein studied is the preprotein, that is, it contains the 26-a.a. mitochondrial localization signal (MLS) sequence. This needs to be clearly stated from the very beginning. With the MLS, the preprotein exits in cytosol not in mitochondria. Also, numbering of the sequence should start at the first residue of the mature form. A sequence alignment including at least *E. coli*, yeast and human mitochondrial chaperonins needs to be presented, and numbering of the Echsp60 secondary structures should follow that of *E. coli* (and human). For example, a3 in the current manuscript should be a4 based on *E. coli* and human chaperonins.
2. The Introduction is very confused with no focuses and has many incorrect descriptions, errors and terminology mix-ups. Introduction should provide general knowledge of chaperonin that is only related to this work, some suggestions are as follow.
 - 1) Information on the two types of chaperonin should be introduced because the protein studied belongs to Type I. Importantly, the protein used in this work has the MLS peptide that is not removed, that is the protein is not a mature protein that is found in mitochondrial matrix. This distinction is important because the potential role of the MLS in chaperonin stability, which is the subject of this work. The inference of this work to the mature form needs to be discussed in Discussion.
 - 2) The main function, to assist protein folding, and its general mechanism need to be included in Introduction. The authors mentioned other roles, apoptosis and inflammation, but what chaperonin does in those biological processes is unclear and may not be related to "maintaining protein homeostasis" (Ln 53). The main function of assisting protein folding, however, is considered "maintaining protein homeostasis".
 - 3) The oligomeric structure of chaperonin (preferably the human mitochondrial chaperonin because they both are type I eukaryotic chaperonins, or the more commonly known *E. coli* GroEL) and the three structural domains of each subunit will need to be clearly described in Introduction because authors are presenting two structures. In particular, the role of the equatorial domain in providing most inter-subunit interactions, i.e., stabilizing the heptameric conformation, needs to be

included, because the domain is reorganized in the dimeric structure.

3. Since the N-terminal residues are involved in forming an inter-subunit β -sheet that is important for the inter-subunit interaction in the heptamer, the MLS peptide which is located directly upstream of the N terminus most likely affects the inter-subunit β -sheet. The authors need to clearly state the differences and relations between the preprotein and mature forms as well as the potential complicated inference from the preprotein form to the functional mature form.

4. The crystal structure of the inactive dimeric mtHsp60 is the most important finding of this work: in particular, the swap of $\alpha 4$ (again was designated as $\alpha 3$ currently) from the original $\alpha 1/\alpha 3/\alpha 4/\alpha 18$ core of its equatorial domain to the neighboring subunit's equatorial domain. The new interface between $\alpha 4$ and two helices of the neighboring subunit is rather extensive and should play an important role in stabilizing the dimer. This new interface is likely extended beyond the currently resolved residues. The N-terminus of $\alpha 4$ is located farthest away from its own subunit so at least some of the missing >100 -residue fragment most likely hangs on somewhere in the neighboring subunit. In addition, since the equatorial domain provides most of the inter-subunit interactions in the active heptamer, both a disruption of the core within the equatorial domain and a formation of new interaction with the neighboring equatorial domain should support strongly that this new interface is important in the dissociation of heptamers to dimers. Thus, this structurally resolved new interface should be analyzed in terms of, for example, the types of interactions and whether the interacting residues are conserved in mitochondrial chaperonins of higher eukaryotic origins (excluding yeast). Such analysis may explain whether such dimer is the common dissociation product among mitochondrial chaperonins and whether such dimer is not expected to form in the bacterial chaperonins. The authors missed this opportunity entirely, but instead, focused on analyzing the 290RLK tripeptide in the apical domain (next).

5. The RLK analysis is not convincing. The authors argued that the RLK motif is conserved in vertebrates, but the motif becomes as KLR in plants and to some extent in bacteria. Such arguments are not convincing. Sequence alignment can only suggest that negatively charged residues, either R or K, are conserved at both 290th and 292nd positions (264th and 266th ?? in the mature protein). Any further interpretations are not convincing. The Ala mutational studies do not provide useful information because they do not mimic bacterial chaperonins which do not form such dimer. The studies using K292A and R290A/K292A mutants should be shortened significantly: for example, deleting Fig. 5a, moving Fig. 5b & c to supplement information. The related discussion section (starting Ln 343 on p15) should be greatly shortened or even deleted.

6. The protein's oligomeric state seems affected by the purification temperature. The full length preprotein mtHsp60 was purified as dimer at 4C but as heptamer at room temperature. Can the purified dimer be converted to the heptamer when incubated at room temperature? Or can the purified heptamer be converted to the dimer when incubated at 4C? The authors also mentioned that the C-terminal truncated mtHsp60 was purified as heptamer at 4C. What was the oligomeric state of the C-terminal truncation if purified at room temperature?

7. Based on the x-ray data table, a 7-fold symmetry, with some extents of restraint, is used in structural refinement of the heptamer. Otherwise, the structure would be over-refined because the number of the unique reflections (52,000) is much below what is required to refine a structure of 400 kDa. Important refinement details such as this need to be disclosed. With C7 imposed in the refinement, comparison of conformation among the seven subunits is not meaningful at all (p 10, also Fig. 4b).

Specific concerns:

Introduction:

Ln 39: "Type I chaperone": should be "Type I chaperonin"

Ln 54: "mtHsp60 is highly conserved": should be "chaperonin is highly conserved"

Ln 63: "during refolding process": should be "in assisting protein folding"

Ln 63: "The equatorial domain forms the active site for ATP binding and hydrolysis which provide the energy for accelerating protein refolding": it is unclear what the authors try to convey. What is important is that the equatorial domain contains the ATP-binding site because this binding site is disrupted in the ATP-deficient dimer. Also, chaperonin does not necessarily "accelerate" protein folding.

Ln 66: "both mtHsp60 and GroEL can self-assemble into a homoheptamer": GroEL is a tetradecamer!

Ln 68: "Hsp60": does it mean mtHsp60?

Ln 73: "mtHsp10 can stabilize the entire complex and regulate...": incorrect statement

Ln 77: the reference of Richardson is incorrect

Ln 79: the reference of Enriquez is incorrect

Ln 103: "an inactive dimeric conformation": should be "a dimer that is deficient in ATPase activity"

Results:

Ln 127 and Fig. 1b: Heptamer will not survive in the denatured SDS gel. You will need to use native gel.

Ln 151 and Fig. 1d: How is the ATPase rate of EcmtHsp60 compared to that of human mHsp60?

Ln 155 and Fig. 1b: Was EcHsp602 purified as a single band on SDS? Was the EcHsp602 sample on the SDS gel freshly prepared or at 4C for some extended time?

Ln 190: "the full-length mtHsp60": this is the preprotein not the mature form

Fig. 3: The structure needs to be annotated with at least the N- and C-termini, the apical, intermediate and equatorial domains, the numbering of the secondary structures. The distances are too small to read - can be listed in a table. The locations of c and d panels in a or b panels should be annotated. Fig. 3d should be better presented. Panel e should be deleted.

Ln 219: Could the extra density be a metal ion, Mg²⁺?

Ln 231: Several terms, mtHsp60, Hsp60, EcHsp60, EcHsp607 etc have been used. Need to keep the term consistent. Also, the C-term truncated version needs to have its own distinction.

Ln 238 and Fig 4b: Was C7 used in refining the heptameric structure? If so, the superposition analysis is not meaningful and Fig. 4b should be deleted.

Fig. 4c: The color contrast for two subunits is too low to distinguish the origins of the b-strands in the inter-subunit b-sheet.

Fig. 4d: Delete the gray subunit. The green subunit of the heptamer should have more structural elements at the N-terminus including the two b-strands involved in inter-subunit interaction.

Ln 251: "all b-sheet structures": name the b strands.

Ln 252: "the missing a1 and a2...": should be "a1 and a2 in the heptamer are not resolved in the dimer".

Ln 258 and Fig. 4e: "while EcHsp602 loses this structure": what do "loses" and "this structure" mean? Fig. 4e is too busy and does not convey any meanings.

Ln 263-296 and Fig. 5: need to greatly tone down, as described in the RLK paragraph.

Discussion:

Ln 309-312: "... the same orientation...": what does the "same" mean? The authors should emphasize on the differences inter-subunit interface (orientation) between the dimer and the heptamer.

Ln 313: Is there any evidence for dissociation of dimer to monomer?

Ln 315: The authors should focus the disruption of the ATP-binding site on the dislocation of the helix (a4) which is the key finding of this work. "Disordered" is vague. The dimer structure clearly lends a structural basis: they seem to miss the power a structure can offer.

Ln 319: "complexes": heptamer?

Ln 324-387: These paragraphs need to be completely revised or deleted.

Reviewer #3 (Comments to the Authors (Required)):

In this paper, the structures of dimeric and heptameric forms of a mitochondrial chaperonin from a marine vertebrate are reported. Nevertheless, there are many problems in other aspects of the work and writing as detailed below.

1. Can the authors rule out formation of mixed oligomers comprising E. coli and grouper subunits? The crystal structures may not reflect the solution situation in this regard.

Perhaps the dimeric complex is formed by E. coli and grouper subunits and is therefore not a true assembly intermediate

2. Line 128 - why would the heptamer be retained in the presence of SDS?

3. Lines 148-150 - Fitting to a substrate inhibition model cannot be achieved with only 2 parameters, Km and Vmax. In addition, a decrease in activity at high ATP concentrations has been associated with nested cooperativity (in the case of GroEL) although for a tetradecamer and substrate inhibition has not been observed. All this warrants a reanalysis, rethink and discussion. One

problem is that the Mg⁺⁺ concentration (1 mM) is not in excess of the high ATP concentrations. Maybe the ATP stocks were not buffered, thereby causing the protein to lose activity at low pH.

4. A dimeric chaperonin structure has been reported before (J Mol Biol. 2011 Sep 16;412(2):192-203). It should be compared to the one determined here.

5. Line 81 - not correct that it has not been seen for GroEL (Cell. 2018 Jan 25;172(3):605-617.e11).

6. It appears that chaperonin purification did not involve any of the steps commonly used (usually in combination) to remove bound polypeptides (e.g. acetone precipitation, ATP wash etc.).

Minor comments:

1. Dimeric, trimeric etc. species are oligomeric and not conformational states and should be referred to as such throughout the paper (e.g. lines 86, 112).

2. The English needs improving throughout. Some examples: line 233 - 'remained' should be 'retained'; line 58- identity cannot be shared; line 40 - change 'new' to 'newly';

3. Line 60 - there is much evidence that GroEL does not accelerate folding but actually retards it (e.g. Biochemistry. 2021 Feb 16;60(6):460-464).

Reviewer #1 (Comments to the Authors (Required)):

This study report the cloning, purification and structural analysis of recombinant mitochondrial Hsp60 from *Epinephelus coioides*. It is demonstrated that under low temperature the recombinant protein is isolated as low molecular weight component while at RT it is purified as heptamer. The system is under equilibrium, as in the presence of ATP it assemble into heptamers. The crystal structure of the dimeric form and the c-terminal truncated form is solved. The report of dimers and heptamers is of high interest, however, some issues need to be addressed before publications

1) Visually the crosslinking results show dimers and monomers - not homogenic dimers. This need to be carefully examined using AUC.

Ans: The presence of monomers in the crosslinking assay may be attributed to the limited concentration of crosslinkers, leading to incomplete crosslinking. However, size-exclusion chromatography data indicated that EcHsp60 formed a homogenous oligomeric state with a single peak. The crosslinking assay was performed to further confirm that the homogenous oligomeric state is dimeric complexes. Furthermore, the crystal structure showed that EcHsp60 assembles into a stable dimer, with a favorable assembly free energy and buried surface area. Hence, we though that these findings provide evidence that EcHsp60 forms dimers. In the future studies, we will further investigate the factors influencing oligomerization of EcHsp60 by using AUC and SAXS. We are grateful for your precious suggestions.

2) Was the his-tag removed at the end?. Do you expect it to interfere with assembly (at

least from the structure of heptamer.

Ans: The C-terminal His-tag was not removed from the recombinant EcHsp60 in this study. We do not yet have evidence to confirm whether or not the His-tag will affect the protein assembly of EcHsp60. However, current biochemical analysis showed that EcHsp60 containing the C-terminal His-tag can still form heptamers, as evidenced by both TEM analysis and crystal structures. The ring-shaped structure of EcHsp60 is similar to the tag-free human mtHsp60. Additionally, the C-terminal Gly-Gly-Met repeats form a disordered structure and have not been observed in any known chaperonin structures to our knowledge. Based on this information, it is believed that the presence of the C-terminal His-tag may not affect the ring assembly. Further investigation into the His-tag free structure of EcHsp60 will be conducted in the near future to address this question.

3) more discussion on differences between subunit contacts in various hsp60 and EC Hsp60.

Ans: Thanks for your suggestion. We have compared the heptameric EcHsp60 with human mtHsp60 and *E. coli* GroEL. Our results showed that the ring-shaped structure of the heptameric EcHsp60 is highly conserved with other well-studied type I chaperonins, with only minor differences in subunit contacts. These findings are depicted in Supplementary Figure S7 and described in detail in the Results section at P13, Ln286. On the other hand, the dimeric EcHsp60 displays unique subunit contacts compared to known chaperonin structures. To further explore this, we compared the subunit contacts of dimeric EcHsp60 with the dimeric Cpn60.2 from *Mycobacterium tuberculosis*. The structural comparison can be found in Supplementary Figure S8 and discussed in the Discussion section at P16, Ln365.

4) If I remember well, structure of dimeric cpn60 from mycobacterium was published in the 90s. Can you see differences?

Ans: Yes, two dimeric structures of Cpn60.2 from *Mycobacterium tuberculosis* have been deposited in the Protein Data Bank (PDB code: 1sjp and 3rtk). We have compared the dimeric MtCpn60.2 to the dimeric EcHsp60 and found that their subunit arrangements, interface areas, and inter-subunit interactions are vastly different. Although the interactions between subunits of MtCpn60.2 are relatively weak, we have thoroughly compared and discussed the structural differences between the two proteins in the Discussion section (P16Line365) and have included a superposition in supplementary Fig. S8. We appreciate your suggestion to improve our manuscript.

5) The term functional is misleading here. It is functional in ATP hydrolysis not in

protein folding (this was not examined). Is it active in protein folding? Do you have in hand purified Hsp10 from EC? If not is it active with Hsp10 from other animals?
Ans: Thank you for your suggestions. We have revised the manuscript to ensure more accurate descriptions of the functions of EcHsp60 and have corrected any misleading statements about its refolding and ATPase functions.

For answering your questions:

1. Despite our efforts to clone various versions of EcHsp10, we were unable to obtain soluble recombinant EcHsp10. The recombinant EcHsp10 tended to aggregate after expression and purification. As mtHsp10 is known to play an essential role in the refolding function of human mtHsp60, our study was unable to examine the protein folding function due to the unavailability of a soluble form of EcHsp10.
2. We have not yet tested the Hsp10 gene from other animals. We appreciate this suggestion and will consider testing the compatibility of human mtHsp10 with EcHsp60 in future studies.

Reviewer #2 (Comments to the Authors (Required)):

The authors (Lai et al.) report two crystal structures of mitochondrial chaperonin mtHsp60 from *Epinephelus coioides* in dimeric and heptameric conformations. mtHsp60 is known to exist as an unstable heptamer which dissociates irreversibly to stable dimers. The structure of the dimer illustrates for the first time how the dimer is assembled and explains why the dimer does not hydrolyze ATP as the heptamer. The work provides important insights into the puzzling heptamer-to-dimer dissociation. However, the manuscript has numerous incorrect and even faulty statements, and needs major revisions. The following concerns need to be addressed.

Major concerns:

1. The protein studied is the preprotein, that is, it contains the 26-a.a. mitochondrial localization signal (MLS) sequence. This needs to be clearly stated from the very beginning. With the MLS, the preprotein exits in cytosol not in mitochondria. Also, numbering of the sequence should start at the first residue of the mature form. A sequence alignment including at least *E. coli*, yeast and human mitochondrial chaperonins needs to be presented, and numbering of the EcHsp60 secondary structures should follow that of *E. coli* (and human). For example, a3 in the current manuscript should be a4 based on *E. coli* and human chaperonins.

Ans: We have clearly distinguished between the EcHsp60 containing the MIS as "naïve

EcHsp60" and the mature EcHsp60 in the revised manuscript. A sequence alignment comparison was made between human mtHsp60, yeast mtHsp60, chloroplast Hsp60, and E. coli GroEL in Supplementary Figure S3. The residues were renumbered to start from the first residue of mature mtHsp60, and the references in the manuscript were updated accordingly. We have also re-calculated the secondary structures using the DSSP method and labeled them in the sequence alignment (Supplementary Figure S3). The reference to the $\alpha 3$ helix was corrected to $\alpha 4$ helix, and the necessary updates were made in the manuscript. However, some differences in secondary structures were noticed between EcHsp60, GroEL, and human mtHsp60. To avoid confusion, the secondary structures were numbered based on the heptameric structure of EcHsp60.

2. The Introduction is very confused with no focuses and has many incorrect descriptions, errors and terminology mix-ups. Introduction should provide general knowledge of chaperonin that is only related to this work, some suggestions are as follow. 1) Information on the two types of chaperonin should be introduced because the protein studied belongs to Type I. Importantly, the protein used in this work has the MLS peptide that is not removed, that is the protein is not a mature protein that is found in mitochondrial matrix. This distinction is important because the potential role of the MLS in chaperonin stability, which is the subject of this work. The inference of this work to the mature form needs to be discussed in Discussion. 2) The main function, to assist protein folding, and its general mechanism need to be included in Introduction. The authors mentioned other roles, apoptosis and inflammation, but what chaperonin does in those biological processes is unclear and may not be related to "maintaining protein homeostasis" (Ln 53). The main function of assisting protein folding, however, is considered "maintaining protein homeostasis". 3) The oligomeric structure of chaperonin (preferably the human mitochondrial chaperonin because they both are type I eukaryotic chaperonins, or the more commonly known E. coli GroEL) and the three structural domains of each subunit will need to be clearly described in Introduction because authors are presenting two structures. In particular, the role of the equatorial domain in providing most inter-subunit interactions, i.e., stabilizing the heptameric conformation, needs to be included, because the domain is reorganized in the dimeric structure.

Ans: Thank you for your insightful comments and the precise suggestions. This work solved the crystal structure of naïve EcHsp60 containing MIS (mitochondria importing signal). It is appropriate to discuss the effect of the MIS on the structure and function of mtHsp60. Hence, we cloned and purified the mature form of EcHsp60 for comparison with the naïve form. Interestingly, both forms of EcHsp60 formed dimers

when purified at low temperature (Fig. 1a). When purified at room temperature, the mature form of EcHsp60 also primarily formed heptamers, with an additional product of larger molecular size (Fig. 1a). We have compared and discussed these findings in the Results section (P6, Ln 115) and the Discussion section (P16, Ln 345). However, these results do not support a correlation between oligomerization and the presence of the MIS. Therefore, we have focused on discussing the subunit interactions in the revised manuscript. Furthermore, in response to your constructive suggestions, the Introduction section has been thoroughly reorganized and rewritten in the revised manuscript (P3, Ln 36).

3. Since the N-terminal residues are involved in forming an inter-subunit β -sheet that is important for the inter-subunit interaction in the heptamer, the MLS peptide which is located directly upstream of the N terminus most likely affects the inter-subunit β -sheet. The authors need to clearly state the differences and relations between the preprotein and mature forms as well as the potential complicated inference from the preprotein form to the functional mature form.

Ans: Thank you for your suggestion. We have constructed the mature form of EcHsp60 and purified the recombinant proteins to study its oligomerization states. Interestingly, both the naïve and mature forms of EcHsp60 tend to form 7-mers when purified at room temperature, while they form dimers when purified at low temperature environments (Fig. 1a). Therefore, we believe that the MLS does not affect the oligomerization of heptameric EcHsp60 or the formation of dimers. We have added these data in Fig. 1 and described the results in Ln111. Both the naïve and mature forms of EcHsp60 are referred to throughout the manuscript. We hope these results will convince audiences that mtHsp60 can form dimers in both its naïve and mature forms.

4. The crystal structure of the inactive dimeric mtHsp60 is the most important finding of this work: in particular, the swap of $\alpha 4$ (again was designated as $\alpha 3$ currently) from the original $\alpha 1/\alpha 3/\alpha 4/\alpha 18$ core of its equatorial domain to the neighboring subunit's equatorial domain. The new interface between $\alpha 4$ and two helices of the neighboring subunit is rather extensive and should play an important role in stabilizing the dimer. This new interface is likely extended beyond the currently resolved residues. The N-terminus of $\alpha 4$ is located farthest away from its own subunit so at least some of the missing >100-residue fragment most likely hangs on somewhere in the neighboring subunit. In addition, since the equatorial domain provides most of the inter-subunit interactions in the active heptamer, both a disruption of the core within the equatorial domain and a formation of new interaction with the neighboring equatorial domain

should support strongly that this new interface is important in the dissociation of heptamers to dimers. Thus, this structurally resolved new interface should be analyzed in terms of, for example, the types of interactions and whether the interacting residues are conserved in mitochondrial chaperonins of higher eukaryotic origins (excluding yeast). Such analysis may explain whether such dimer is the common dissociation product among mitochondrial chaperonins and whether such dimer is not expected to form in the bacterial chaperonins. The authors missed this opportunity entirely, but instead, focused on analyzing the 290RLK tripeptide in the apical domain (next).

Ans: Thank you for your feedback and suggestions. We fully agree with your viewpoint that the swapped $\alpha 4$ helix plays a crucial role in the dimeric interaction. Our analysis of the interactions between the two protomers of the dimeric EcHsp60 has emphasized the $\alpha 4$ helix, due to its large interface. However, the $\alpha 4$ helix primarily interacts with another subunit through hydrophobic interactions and van der Waals forces, with only two polar contacts observed at each $\alpha 4$ helix. The interacting residues, T91, E100, R444, and D494, are highly conserved among GroEL and mtHsp60, with only D103 not being conserved in chaperonins, making it challenging to identify a specific target for subsequent mutagenesis studies.

In contrast, the RLK motif formed four hydrogen bonds between two subunits of the dimeric EcHsp60, which are important for stabilizing the dimer. The RLK motif is only observed in animal mtHsp60, while the KLR motif is conserved in yeast, plant mtHsp60, and some bacterial GroEL. The RLK motif forms hydrogen bonds but not salt bridges to form the dimeric structure, so the positive charge is not necessary for maintaining dimeric interactions. In contrast, the side chain of R264 forms inter- and intra-molecular hydrogen-bond networks, which cannot be formed by lysine residues. Hence, we proposed that the RLK tripeptide is crucial for stabilizing the dimeric conformation of mtHsp60. We have added these descriptions in the manuscript at P10Line223 to fully explain our rationale for focusing on the RLK motif. Additionally, we have greatly expanded the descriptions and discussions on the $\alpha 4$ helix interface in the revised manuscript. We hope that these modifications will convince the audience of the importance of both the $\alpha 4$ helix and RLK motif in forming the dimeric mtHsp60 in animals.

5. The RLK analysis is not convincing. The authors argued that the RLK motif is conserved in vertebrates, but the motif becomes as KLR in plants and to some extent in bacteria. Such arguments are not convincing. Sequence alignment can only suggest that negatively charged residues, either R or K, are conserved at both 290th and 292nd

positions (264th and 266th ?? in the mature protein). Any further interpretations are not convincing. The Ala mutational studies do not provide useful information because they do not mimic bacterial chaperonins which do not form such dimer. The studies using K292A and R290A/K292A mutants should be shortened significantly: for example, deleting Fig. 5a, moving Fig. 5b & c to supplement information. The related discussion section (starting Ln 343 on p15) should be greatly shortened or even deleted.

Ans: We appreciate your input and understand the importance of presenting the results accurately. We have taken your feedback into consideration and revised the manuscript accordingly. We have deleted Figure 5a and greatly reduced the discussion on the importance of the RLK motif. However, as mentioned in last question, the crystal structure demonstrated that the RLK motif contributes essential binding forces to stabilize the dimeric interaction. The alanine mutagenesis also provides evidence that altering residue bonding of RLK motifs will change its oligomeric state. These experimental results are repeatable and reliable. They should be revealed to study the functional role of this RLK motif. Therefore, we have kept Figure 5 in the revised manuscript while avoiding over-explanation of the current results.

6. The protein's oligomeric state seems affected by the purification temperature. The full length preprotein mtHsp60 was purified as dimer at 4C but as heptamer at room temperature. Can the purified dimer be converted to the heptamer when incubated at room temperature? Or can the purified heptamer be converted to the dimer when incubated at 4C? The authors also mentioned that the C-terminal truncated mtHsp60 was purified as heptamer at 4C. What was the oligomeric state of the C-terminal truncation if purified at room temperature?

Ans:

1. The purified dimer cannot be converted to a heptamer without ATP. The current evidence shows that ATP is essential for inducing the reassembly. The control group in Fig. 2a showed that the dimeric EcHsp60 did not convert into heptamers.
2. The heptameric naïve EcHsp60 cannot be converted to dimers when incubated at 4°C. We tried to incubate the heptamers at 4°C for 2 hours and analyzed by size-exclusion chromatography, and the heptameric peak did not change significantly. Therefore, we think that temperature is not sufficient to convert the oligomeric states of EcHsp60.
3. Yes, the C-terminal truncated mtHsp60 was also purified as heptamers at room temperature. The data is added in the Supplementary S6a.

7. Based on the x-ray data table, a 7-fold symmetry, with some extents of restrain, is

used in structural refinement of the heptamer. Otherwise, the structure would be over-refined because the number of the unique reflections (52,000) is much below what is required to refine a structure of 400 kDa. Important refinement details such as this need to be disclosed. With C7 imposed in the refinement, comparison of conformation among the seven subunits is not meaningful at all (p 10, also Fig. 4b).

Ans: Thanks for your comments. Due to the lower resolution of the diffraction data (below 3Å), we applied “secondary structure restraint” and “Ramachandra restraint” to maintain the structural properties during refinement. However, NCS restraints were not utilized for the refinement of the heptameric EcHsp60. Instead, we employed TLS restraint refinement to reduce the Rfree value. The refinement details have been added to the Materials and Methods section on P23, Ln528. Additionally, the comparison of the seven protomers in the heptamer has been removed from Fig. 4b.

Specific concerns:

Introduction:

Ln 39: "Type I chaperone": should be "Type I chaperonin"

Ans: Thank you for the correction. The original sentence has been deleted, and we have checked that this typo does not appear in the revised version.

Ln 54: "mtHsp60 is highly conserved": should be "chaperonin is highly conserved"

Ans: The original sentence has been deleted. Thanks for the comment.

Ln 63: "during refolding process": should be "in assisting protein folding"

Ans: We thought that this modification is more accurate. The sentence is corrected in the revised manuscript at P4, Ln74.

Ln 63: "The equatorial domain forms the active site for ATP binding and hydrolysis which provide the energy for accelerating protein refolding": it is unclear what the authors try to convey. What is important is that the equatorial domain contains the ATP-binding site because this binding site is disrupted in the ATP-deficient dimer. Also, chaperonin does not necessarily "accelerate" protein folding.

Ans: Thanks for your correction. This sentence is modified in the revised manuscript at P4, Ln70.

Ln 66: "both mtHsp60 and GroEL can self-assemble into a homoheptamer": GroEL is a tetradecamer!

Ans: The description about GroEL is deleted in the paragraph at P3, Ln54.

Ln 68: "Hsp60": does it mean mtHsp60?

Ans: Thanks for your correction. The word is modified in the revised manuscript at P3, Ln56.

Ln 73: "mtHsp10 can stabilized the entire complex and regulate...": incorrect statement

Ans: We apology for this mistake. This sentence is deleted in the revised manuscript.

Ln 77: the reference of Richardson is incorrect

Ans: Thanks for your correction. The paragraph in Introduction is revised and thus the reference is deleted.

Ln 79: the reference of Enriquez is incorrect

Ans: Thanks for your correction. The paragraph in Introduction is revised and thus the reference is deleted.

Ln 103: "an inactive dimeric conformation": should be "a dimer that is deficient in ATPase activity"

Ans: Thanks for your correction. The "inactive" is not an accurate description. The sentence is corrected at P5, Ln97.

Results:

Ln 127 and Fig. 1b: Heptamer will not survive in the denatured SDS gel. You will need to use native gel.

Ans: our experimental results showed that heptameric naïve EcHsp60 remained weak signals in SDS-PAGE and Western blot. All samples are boiling in the sample buffer containing SDS and 2-Me for 10 minutes. We thought that these signals is represent the oligomeric EcHsp60 that are not fully denatured. Interestingly, these signals are not detected in mature EcHsp60 heptamers. It is not clear how MIS affecting the oligomerization. The oligomeric states of both naïve and mature EcHsp60 are studied by using size exclusion chromatography. To avoid misleading, we removed the description about the undenatured proteins in SDS-PAHE in the revised manuscript. Further studies will be continued to answer the reasons for the SDS resistance of heptamer naïve EcHsp60.

Ln 151 and Fig. 1d: How is the ATPase rate of EcmtHsp60 compared to that of human mHsp60?

Ans: We have reviewed the references and found that the ATPase activity of mtHsp60/mtHsp10 has been reported to range from 0.75 nmol/min to 0.05 μ M/min in different studies (*American Journal of Molecular Biology*, 2012, 2, 93-102 and *Scientific Reports*, 2017, 7, 16931). The wide variation in the reported ATPase activity could be due to differences in the quantification methods used. As a result, we have decided not to compare the ATPase activity of EcHsp60 with that of human Hsp60 in this study.

Ln 155 and Fig. 1b: Was EcHsp60 purified as a single band on SDS? Was the EcHsp60 sample on the SDS gel freshly prepared or at 4°C for some extended time?

Ans: The EcHsp60 can be purified as a major band on SDS-PAGE and degraded rapidly when stored at 4°C. The samples for SDS-PAGE are freshly prepared after purification and boiled for 10 min before storing at -20°C. Current data (Fig. 1b) showed that naïve EcHsp60 seems more sensitive compared to mature EcHsp60. However, more experimental evidence is required for understanding the mechanisms.

Ln 190: "the full-length mtHsp60": this is the preprotein not the mature form

Ans: The original sentence has been removed in the revised manuscript. We have clearly distinguished the naïve EcHsp60 from the mature EcHsp60 in the other sections.

Fig. 3: The structure needs to be annotated with at least the N- and C-termini, the apical, intermediate and equatorial domains, the numbering of the secondary structures. The distances are too small to read - can be listed in a table. The locations of c and d panels in a or b panels should be annotated. Fig. 3d should be better presented. Panel e should be deleted.

Ans: Thanks for the suggestions for improving the figure quality. The Fig 3 is modified according to your great suggestions. The polar interactions are listed in Supplementary table S3. All figures are labeled with more detailed annotations for better reading.

Ln 219: Could the extra density a metal ion, Mg²⁺?

Ans: The extra density in the Fo-Fc map is higher than 10 sigma and too large to fit a magnesium ion. While magnesium typically forms octahedral coordination with surrounding oxygen molecules, no contact residues were identified to provide such interactions. Therefore, it is unlikely that the extra density represents a metal ion.

Ln 231: Several terms, mtHsp60, Hsp60, EcHsp60, EcHsp607 etc have been used. Need to keep the term consistent. Also, the C-term truncated version needs to have its own distinction.

Ans: Thank you for your suggestions. We have checked these terms through the manuscripts, and termed the dimeric EcHsp60 as EcHsp60₂, heptameric EcHsp60 as EcHsp60₇. The C-terminal truncated EcHsp60 is also termed EcHsp60^{AC} to avoid the confusion during reading.

Ln 238 and Fig 4b: Was C7 used in refining the heptameric structure? If so, the superposition analysis is not meaningful and Fig. 4b should be deleted.

Ans: Thank you for pointing this out. We have verified the refinement procedures used for the heptameric EcHsp60 structure and confirm that secondary structure restraints, Ramachandran restraints, and TLS restraints refinements were applied. We did not utilize NCS refinement to impose C7 symmetry during the refinement process. However, we realized that Fig. 4b only provided limited information for discussion of the heptameric structures. As such, we have replaced Fig. 4b with a figure that displays the distribution of secondary structures in the protomers of the EcHsp60 heptamers.

Fig. 4c: The color contrast for two subunits is too low to distinguish the origins of the b-strands in the inter-subunit b-sheet.

Ans: The colors of Fig.4c were changed to green and purple and the viewing angle was also adjusted for better distinguish the b-strands.

Fig. 4d: Delete the gray subunit. The green subunit of the heptamer should have more structural elements at the N-terminus including the two b-strands involved in inter-subunit interaction.

Ans: Thank you for pointing that out. We have corrected our mistake and now clearly describe the gray subunit in the revised figures (Fig. 4d and 4e) to better exhibit the structural comparison and the replacement of $\alpha 4$ helix in the equatorial domain, respectively.

Ln 251: "a1 b-sheet structures": name the b strands.

Ans: Thank you for your reminder. We used an incorrect term to describe the secondary structure. All "b-sheet" terms have been corrected in the revised manuscripts.

Ln 252: "the missing a1 and a2...": should be "a1 and a2 in the heptamer are not resolved in the dimer".

Ans: Thank you for your correction. The term "missing" is not accurate to describe the

disordered regions in the crystal structures. The original sentence has been replaced at P12, Ln271, and we have corrected this term throughout the revised manuscripts.

Ln 258 and Fig. 4e: "while EcHsp602 loses this structure": what do "loses" and "this structure" mean? Fig. 4e is too busy and does not convey any meanings.

Ans: This term is not accurate in describing the disruption of the ATP binding pocket through the rearrangement of the equatorial domain. Therefore, we have completely revised this paragraph and created new figures (Fig. 4d and 4e) to clearly illustrate the conformational changes in the equatorial domain and highlight the disruption of the ATP binding pockets. These revisions can be found on P12, Ln274.

Ln 263-296 and Fig. 5: need to greatly tone down, as described in the RLK paragraph.

Ans: As previously stated, we agree that comparing the prokaryotic GroEL to animal mtHsp60 may not be convincing. Therefore, we have removed Fig. 5a and the discussion regarding the difference between the RLK and KLR motifs. Despite this, our biochemical evidence shows that the double mutants formed mostly ring-shaped heptamers when purified at low temperature, providing solid experimental results. We have retained partial descriptions to discuss the functional importance of the RLK in stabilizing the dimeric interaction.

Discussion:

Ln 309-312: "... the same orientation...": what does the "same" mean? The authors should emphasize on the differences inter-subunit interface (orientation) between the dimer and the heptamer.

Ans: In this section, we attempt to discuss the difference in subunit arrangement between the dimer and heptamer. We propose that the assembled orientation of the dimer is distinct from that of the heptamer, and therefore the dimers cannot be directly packed into heptamers without dissociation. This revised information can be found in the manuscript on P16, Ln334.

Ln 313: Is there any evidence for dissociation of dimer to monomer?

Ans: This speculation is based on the observation that dimers can be reconstituted into heptamers after incubation with ATP at 37°C (Fig. 2). If the dimers did not dissociate into monomers, then they could not be assembled into heptamers as their subunit interfaces would be occupied.

Ln 315: The authors should focus the disruption of the ATP-binding site on the dislocation of the helix (a4) which is the key finding of this work. "Disordered" is vague.

The dimer structure clearly lends a structure basis: they seem to miss the power a structure can offer.

Ans: Thanks for your valuable suggestions. Thank you for your suggestion. We have revised the figure to clearly illustrate the disruption of the ATP-binding pocket in the dimeric EcHsp60 (Fig. 4e). We believe that this updated description will help the audience better understand how the dimers lost their ATPase activity.

Ln 319: "complexes": heptamer?

Ans: Yes, these studies proposed ATP-induced ring assembly and disassembly. The sentence is corrected at P16, Ln341.

Ln 324-387: These paragraphs need to be completely revised or deleted.

Ans: Thank you for your suggestion. We have streamlined the Discussion section to focus on the structural comparisons with dimeric MtCpn60.2 and avoided over-explanation. This revision has helped to clarify the key findings and maintain the focus of the discussion on the main results.

Reviewer #3 (Comments to the Authors (Required)):

In this paper, the structures of dimeric and heptameric forms of a mitochondrial chaperonin from a marine vertebrate are reported. Nevertheless, there are many problems in other aspects of the work and writing as detailed below.

1. Can the authors rule out formation of mixed oligomers comprising *E. coli* and grouper subunits? The crystal structures may not reflect the solution situation in this regard. Perhaps the dimeric complex is formed by *E. coli* and grouper subunits and is therefore not a true assembly intermediate

Ans: We have three reasons to explain the formation of the dimeric complex by grouper mtHsp60 without the involvement of *E. coli* GroEL.

1. Previous studies have expressed human mtHsp60 in *E. coli* and obtained highly pure protein for structural analysis, however, there is no evidence of eukaryotic mtHsp60 forming oligomers with prokaryotic GroEL (*Proc Natl Acad Sci U S A*, 2015,112(19): 6044-6049; *Nat Commun*, 2002, 11(1): 1916; *iScience*, 2021, 24(1): 102022). Furthermore, the molecular size of the naïve EcHsp60 (61.2 kDa) is slightly larger than the mature EcHsp60 (58.3 kDa) and *E. coli* GroEL (57.3 kDa) (as shown in Figure 1b). If *E. coli* GroEL had formed a complex with EcHsp60, it should have been separable from the naïve EcHsp60 in SDS-PAGE and not detected in the Western blot. However, in Figure 1b, both SDS-PAGE and Western blot

shows a pure band in naïve EcHsp60, indicating that *E. coli* GroEL does not oligomerize with EcHsp60.

2. The Western blot showed similar signals for the same amounts of heptameric and dimeric EcHsp60 in SDS-PAGE. If the dimeric EcHsp60 was composed of *E. coli* GroEL and EcHsp60, the Western blot (anti-his tag) signals should have been lower than heptameric EcHsp60 in the same loading protein amounts.
3. The resolution of the dimeric EcHsp60 structure was determined to be 2.35 Å, and the side chains could be clearly observed and traced in the density map. The b-factors and occupancies are similar between chain A and chain B, providing structural evidence that the dimeric complex is formed by two EcHsp60 monomers. Although the dimeric conformation may be a result of crystal packing, the PISA program showed that the two subunits in the current structure form a stable dimer with a buried surface and free energy that meet the dimerization criteria (Supplementary Fig. S2).

In conclusion, our experimental results suggest that the EcHsp60 forms dimers independently without the involvement of *E. coli* GroEL.

2. Line 128 - why would the heptamer be retained in the presence of SDS?

Ans: Our observations indicate that some protein complexes with strong inter-subunit interactions may not be fully denatured during SDS-PAGE analysis. In this case, the naïve EcHsp60 were treated with SDS and boiled for ten minutes. However, it still showed oligomeric bands in the SDS-PAGE and Western blot (Fig. 1b). However, mature EcHsp60 did not to form SDS-resistant oligomers despite having similar SEC profiles as the naïve form. The mechanism behind the effect of the N-terminal MIS on complex stability is not yet clear. To prevent confusion, the description of retained oligomers has been deleted from the revised manuscript. Instead, we believe that the negative stain TEM results provide strong evidence that EcHsp60 forms heptameric complexes when purified at room temperature.

3. Lines 148-150 - Fitting to a substrate inhibition model cannot be achieved with only 2 parameters, K_m and V_{max} . In addition, a decrease in activity at high ATP concentrations has been associated with nested cooperativity (in the case of GroEL) although for a tetradecamer and substrate inhibition has not been observed. All this warrants a reanalysis, rethink and discussion. One problem is that the Mg^{++} concentration (1 mM) is not in excess of the high ATP concentrations. Maybe the ATP stocks were not buffered, thereby causing the protein to lose activity at low pH.

Ans: Thank you for your input. We have conducted the ATPase activity assay multiple

times in our lab and obtained consistent results. The assay medium was buffered with 50 mM Tris-HCl at pH 7.5, and the ATP concentration of 3.2 mM did not significantly affect the pH. Additionally, the ATPase activity decreased when the ATP concentration was raised to 400 μ M in the presence of sufficient Mg^{2+} ions (1 mM). This suggests that substrate inhibition may be a real effect for heptameric EcHsp60, although the underlying mechanism is not yet clear. Our main focus in presenting this data was to demonstrate that heptameric EcHsp60 is active in ATP hydrolysis, while dimeric EcHsp60 is inactive. Therefore, we adjusted the substrate concentration range to 0.3-200 μ M and updated the corresponding descriptions in the revised manuscript (P7, Ln 146). We hope to further investigate and understand the substrate inhibition effect of EcHsp60 in future studies.

4. A dimeric chaperonin structure has been reported before (J Mol Biol. 2011 Sep 16;412(2):192-203). It should be compared to the one determined here.

Ans: Thank you for your suggestion. We have revised the description and included the comparison between dimeric EcHsp60 and dimeric MtCpn60.2 in the Discussion section at P15Line365, highlighting the structural differences between the two dimers. The comparison is also visualized in the supplementary figure S8, which provides a clear comparison of the subunit arrangement and interactions. This information provides a better understanding of the structural differences between these two dimeric chaperonins.

5. Line 81 - not correct that it has not been seen for GroEL (Cell. 2018 Jan 25;172(3):605-617.e11).

Ans: Thanks for your correction. We have deleted this description and rewrite the paragraphs in the revised manuscript to avoid misleading.

6. It appears that chaperonin purification did not involve any of the steps commonly used (usually in combination) to remove bound polypeptides (e.g. acetone precipitation, ATP wash etc.)

Ans: In our purification procedures, we did not take specific steps to remove bound polypeptides, as we found that the purity of eluted EcHsp60 was over 90% (as shown in Fig. 1), which is suitable for biochemical and structural analysis. Recent cryo-EM studies have shown that human mtHsp60 can be expressed in *E. coli* and purified using a Ni-NTA column without further washing steps (*Nat Commun*, 2002, 11(1): 1916; *iScience*, 2021, 24(1): 102022). Thus, we believe that removing bound polypeptides may not be necessary when expressing eukaryotic mtHsp60 in a prokaryotic system, given the absence of mtHsp10 in the environment.

Minor comments:

1. Dimeric, trimeric etc. species are oligomeric and not conformational states and should be referred to as such throughout the paper (e.g. lines 86, 112).

Ans: Thank you for the comments. We have made the necessary corrections and revised the terms to reflect the accurate descriptions of oligomeric states in the manuscript.

2. The English needs improving throughout. Some examples: line 233 - 'remained' should be 'retained'; line 58- identity cannot be shared; line 40 - change 'new' to 'newly';

Ans: Thanks for your precise suggestion. We have corrected these mistakes and revised the English throughout all the manuscripts. The modified regions are marked in blue in the revised manuscripts.

3. Line 60 - there is much evidence that GroEL does not accelerate folding but actually retards it (e.g. Biochemistry. 2021 Feb 16;60(6):460-464).

Ans: Thanks very much for your correction. The original sentence is not accurate, and we have deleted it in the revised manuscripts.

March 7, 2023

Re: Life Science Alliance manuscript #LSA-2022-01753-TR

Dr. Shih-Ming Lin
National Cheng Kung University
Department of Biotechnology and Bioindustry Sciences
1, University Rd,
East Dist.,
Tainan city, Tainan 701
Taiwan

Dear Dr. Lin,

Thank you for submitting your revised manuscript entitled "Crystal structures of dimeric and heptameric mtHsp60 reveal the mechanism of chaperonin inactivation" to Life Science Alliance. The manuscript has been seen by the original reviewers whose comments are appended below. While the reviewers continue to be overall positive about the work in terms of its suitability for Life Science Alliance, some important issues remain.

Our general policy is that papers are considered through only one revision cycle; however, given that the suggested changes are relatively minor, we are open to one additional short round of revision. Please note that I will expect to make a final decision without additional reviewer input upon re-submission.

Please submit the final revision within one month, along with a letter that includes a point by point response to the remaining reviewer comments.

To upload the revised version of your manuscript, please log in to your account: <https://lsa.msubmit.net/cgi-bin/main.plex>
You will be guided to complete the submission of your revised manuscript and to fill in all necessary information.

B. MANUSCRIPT ORGANIZATION AND FORMATTING:

Sincerely,

Reviewer #2 (Comments to the Authors (Required)):

The manuscript has improved substantially in terms of content and writing, however, much still needs to be revised. Many references are either outdated or inaccurately cited.

The following are some examples of inaccurate statements.

Ln 21-22: Whether mtHsp60 functions as heptamer or tetradecamer is still debatable. What is known is that it exists as a heptamer.

Ln 43: change "Mitochondria" to "Mitochondrial"

Ln 57: incorrectly cited reference, Braig et al, 1994 is a GroEL paper

Ln 58-60: incorrect statement. The association may occur in the presence of mtHsp10 and ATP.

Ln 58-60: change "forma" to "forms a "

Ln 58-68: please revise the sentences; duplicated and conflicting sentences

Ln 70: The Braig 1994 and Xu 1997 references are on GroEL, not mtHsp60.

Ln 70-76: By convention, there are no E1 and E2 subdomains, instead, it is the equatorial domain. Similarly, no I1 and I2 subdomains, it is just the intermediate domain.

Ln 74-75: is Ishida a structure paper? Also, references are not necessary here.

Ln 78: delete the references

Ln 93-94: Not entirely correct. A structural basis has been proposed for the heptameric instability, please include it here.

173-174: The monomer to heptamer reconstitution was used to purify the heptameric mHsp60, and such purification was used in the referenced papers. However, these papers are not the first to report such purification. Need to cite the original paper, Viitanen et al 1992 or 1998

Ln 225-234: The description of the 264RLK266 interactions is very confusing. Explain "The RLK motif forms hydrogen bonds, but not salt bridges": does it mean that the interactions involve atoms from the backbone? Also, explain which h-bonds cannot be established in the sentence "side chain of R264 forms... cannot be established by lysine". When the KLR sequence is modeled into the RLK structure, what are the four intermolecular distances in Table S3? I would expect similar results. Basically, I am not convinced that the RLK motif (in the animal) is unique from the KLR motif (in the plant, etc). The conclusion and extrapolation are not solid.

Ln 240: "may play a role in regulating the formation of dimeric EcHsp60"; delete "regulating

Ln 247-249: Where, what residue, is the truncation? State explicitly. Also, need to include the reference for human mtHsp60.

Ln 282: The Kim 1994 citation is incorrect as it discusses type II chaperonin not type I human mtHsp60.

Ln 307: What are the differences between "vertebrate" and "animal"? Also, what is the percentage of conservation by "highly conserved"? A reference is required too.

Ln 246-348: Incorrect statements. It is not "nucleotide binding and release", rather, it is the presence of nucleotide particularly ATP. Poor references. The cited references are mixed with research and review papers. And some of the cited references are not the first reports.

Ln 358: "luminal side"? Inside the folding chamber? If so, the folding chamber needs to be introduced.

Ln 366: Check the reference.

Ln 371-411: Condense the comparison with MtCpn60.2.

Ln 412-420: Delete. Results of the in vitro trypsin digestion cannot be extrapolated to in cell protein metabolism situation.

Ln 421-433: Delete. The dimeric EcmtHsp60 is a product obtained at 4C but it is not known without any evidence whether it bears any biological and physiological relevance.

Figures 3 and 4 are reversed.

Reviewer #3 (Comments to the Authors (Required)):

The revised version is improved but some issues remain.

1. Lines 152-154 - The ATPase activity shows no cooperativity? Hard to believe as most chaperonin rings display positive cooperativity. It seems that only one time point (30 min) was measured for each ATP concentrations so the y-axis is not initial velocity. In all likelihood, the y-axis for the low ATP concentrations corresponds to the amount of ATP added.
2. Line 234 - how do the authors distinguish between hydrogen bonds and salt-bridges? Hydrogen bonds between side-chains of E212 and R264 or K266 are salt-bridges.
3. There are still multiple spelling and grammar errors (e.g. 'strains' in line 399).

Reviewer #2 (Comments to the Authors (Required)):

The manuscript has improved substantially in terms of content and writing, however, much still needs to revise. Many references are either outdated or inaccurately cited.

The following are some examples of inaccurate statements.

Ln 21-22: Whether mtHsp60 functions as heptamer or tetradecamer is still debatable. What is known is that it exists as a heptamer.

Ans: Thank you for the suggestion. This sentence is modified as “mtHsp60 self-assembly into a ring shaped heptamer, two of which can further form a double ring tetradecamer in presence of ATP and mtHsp10.” to describe the oligomerization states of mtHsp60 more accurately. This modification is shown on Page 2, Line 15 in the revised manuscript.

Ln 43: change "Mitochondria" to "Mitochondrial"

Ans: We have modified this sentence as suggested on Page 3, Line 37 in the revised manuscript.

Ln57: incorrectly cited reference, Braig et al, 1994 is a GroEL paper

Ans: Thanks for pointing out this mistake. We have removed the incorrect reference from the manuscript and have checked all references to ensure accurate citations. The revised manuscript highlights these changes in blue.

Ln 58-60: incorrect statement. The association may occur in the presence of mtHsp10 and ATP.

Ans: The sentence has been revised as " In the presence of mtHsp10 and ATP, these heptamers could stack back-to-back to form a double-ring tetradecamer " on Page 3, Line 51.

Ln 58-60: change "forma" to "forms a "

Ans: This word has been deleted because the sentence has been modified in the revised manuscript.

Ln 58-68: please revise the sentences; duplicated and conflicting sentences

Ans: Thank for your suggestions. We have deleted the unnecessary statements in the paragraph. The revised paragraph can be found on Page 3, Line 51 in the revised manuscript.

Ln70: The Braig 1994 and Xu 1997 references are on GroEL, not mtHsp60.

Ans: We have revised the sentence to reflect the information from the crystal structure study of mtHsp60 on Page 4, Line 57 and removed unrelated citations from the revised manuscript.

Ln 70-76: By convention, there are no E1 and E2 subdomains, instead, it is the equatorial domain. Similarly, no I1 and I2 subdomains, it is just the intermediate domain.

Ans: We have removed the descriptions mentioned E1, E2, I1 and I2, as suggested in the revised manuscript.

Ln 74-75: is Ishida a structure paper? Also, references are not necessary here.

Ln 78: delete the references

Ans: Thank you for your suggestions. We have removed the unnecessary citations.

Ln 93-94: Not entirely correct. A structural basis has been proposed for the heptameric instability, please include it here.

Ans: Thank you for your feedback. We have removed the incorrect sentence and added the proposed structural basis for the heptameric instability as suggested. The revised paragraph can be found on Page 4, Line 74 of the revised manuscript.

173-174: The monomer to heptamer reconstitution was used to purify the heptameric mHsp60, and such purification was used in the referenced papers. However, these papers are not the first to report such purification. Need to cite the original paper, Viitanen et al 1992 or 1998

Ans: Thank you for pointing this out. We have revised the reference in the revised manuscript on Page 8, Line 152. We appreciate your suggestion in improving the accuracy of our references.

Ln 225-234: The description of the 264RLK266 interactions is very confusing. Explain "The RLK motif forms hydrogen bonds, but not salt bridges": does it mean that the interactions involve atoms from the backbone? Also, explain which h-bonds cannot be established in the sentence "side chain of R264 forms... cannot be established by lysine". When the KLR sequence is modeled into the RLK structure, what are the four

intermolecular distances in Table S3? I would expect similar results. Basically, I am not convinced that the RLK motif (in the animal) is unique from the KLR motif (in the plant, etc). The conclusion and extrapolation are not solid.

Ans: Thank you for your feedback. We apologize for any confusion caused by our previous descriptions and have taken steps to simplify the descriptions on the RLK motif. We have removed unnecessary sentences and revised the paragraph to clarify the structural importance of the RLK motif. The modified paragraph can be found on Page 9, Line 195-200.

Ln 240: "may play a role in regulating the formation of dimeric EcHsp60"; delete "regulating"

Ans: Thank you for your suggestion. We have removed the word "regulating" from the sentence. The modified sentence is now shown on Page 10, Line 199.

Ln 247-249: Where, what residue, is the truncation? State explicitly. Also, need to include the reference for human mtHsp60.

Ans: Thank you for your suggestion. We have made the modification to explicitly state the residue range of the truncation as $\Delta 527-552$. Additionally, we have added the reference for human mtHsp60 on Page 10, Line 214-215 of the revised manuscript.

Ln 282: The Kim 1994 citation is incorrect as it discusses type II chaperonin not type I human mtHsp60.

Ans: Thank you for pointing out this mistake. We have moved this sentence to introduction and replaced this incorrect citation with two more appropriate citations, as shown on Page 4, Line 65 of the revised manuscript.

Ln 307: What are the differences between "vertebrate" and "animal"? Also, what is the percentage of conservation by "highly conserved"? A reference is required too.

Ans: As the descriptions regarding the RLK motif were simplified in the revised manuscript, we have also removed this sentence to avoid over-explaining the role of the motif. The revised paragraph is shown on Page 12, Line 269.

Ln 346-348: Incorrect statements. It is not "nucleotide binding and release", rather, it is the presence of nucleotide particularly ATP. Poor references. The cited references are mixed with research and review papers. And some of the cited references are not the first reports.

Ans: Thank you for your feedback. We found that the sentence is not necessary for explaining out proposed models. Thus, the sentence is removed in the revised manuscripts.

Ln 358: "luminal side"? Inside the folding chamber? If so, the folding chamber needs to introduce.

Ans: Thank you for your suggestion. We have revised the sentences based on your feedback. The revised manuscript will reflect this change on Page 15, Line 322.

Ln 366: Check the reference.

Ans: We have removed the sentence from the revised manuscript since it is not necessary to explain the existence of octameric mtHsp60.

Ln 371-411: Condense the comparison with MtCpn60.2.

Ans: Thank you for your suggestion. We have carefully reviewed and revised the relevant paragraphs based on your feedback. The redundant descriptions have been removed, and the comparison with MtCpn60.2 has been condensed into one paragraph on Page 15, Line 328 in the revised manuscript. We appreciate your input in helping us improve the clarity and conciseness of our manuscript.

Ln 412-420: Delete. Results of the in vitro trypsin digestion cannot be extrapolated to in cell protein metabolism situation.

Ln 421-433: Delete. The dimeric EcmtHsp60 is a product obtained at 4C but it is not known without any evidence whether it bears any biological and physiological relevance.

Ans: Based on your suggestion, we have removed the paragraphs to avoid over-explanation. Instead, we have added a summary in the last paragraph to conclude our findings in this study. The summary paragraph is shown on Page 16, Line 356.

Figures 3 and 4 are reversed.

Ans: We apologize for the mistake made during the figure upload process. We will correct this mistake when submitting the revised manuscript.

Reviewer #3 (Comments to the Authors (Required)):

The revised version is improved but some issues remain.

1. Lines 152-154 - The ATPase activity shows no cooperativity? Hard to believe as

most chaperonin rings display positive cooperativity. It seems that only one time point (30 min) was measured for each ATP concentrations so the y-axis is not initial velocity. In all likelihood, the y-axis for the low ATP concentrations corresponds to the amount of ATP added.

Ans: Thank you for your suggestion. We have recalculated the kinetic results using an allosteric sigmoid function and found that the R-squared value improved from 0.97 to 0.98. The Hill slopes of both curves are greater than 1, indicating positive cooperativity. We have corrected the results on Page 7, Line 129, and included a table showing the kinetic values in Supplementary Table S1. Your correction has greatly improved the accuracy in interpreting our results and we appreciate your feedback.

2. Line 234 - how do the authors distinguish between hydrogen bonds and salt-bridges? Hydrogen bonds between side-chains of E212 and R264 or K266 are salt-bridges.

Ans: We have removed these confusing statements and expectations in the revised manuscripts to improve clarity. The modified paragraph can be found on Page 9, Line 195.

3. There are still multiple spelling and grammar errors (e.g. 'strains' in line 399).

Ans: Thank you for your feedback. We have thoroughly checked for spelling and grammar errors in the revised manuscript and have made corrections wherever necessary. These changes have been highlighted in blue in the revised manuscript.

March 14, 2023

RE: Life Science Alliance Manuscript #LSA-2022-01753-TRR

Dr. Shih-Ming Lin
National Cheng Kung University
Department of Biotechnology and Bioindustry Sciences
1, University Rd,
East Dist.,
Tainan city, Tainan 701
Taiwan

Dear Dr. Lin,

Thank you for submitting your revised manuscript entitled "Crystal structures of dimeric and heptameric mtHsp60 reveal the mechanism of chaperonin inactivation". We would be happy to publish your paper in Life Science Alliance pending final revisions necessary to meet our formatting guidelines.

- please add the Twitter handle of your host institute/organization as well as your own or/and one of the authors in our system
- please upload your supplementary figures as single files
- please add a conflict of interest statement to your main manuscript text
- please consult our manuscript preparation guidelines <https://www.life-science-alliance.org/manuscript-prep> and make sure your manuscript sections are in the correct order
- please add a figure callout for Figure S7a,b and S7d,e to the main manuscript text

A. FINAL FILES:

B. MANUSCRIPT ORGANIZATION AND FORMATTING:

Sincerely,

March 16, 2023

RE: Life Science Alliance Manuscript #LSA-2022-01753-TRRR

Dr. Shih-Ming Lin
National Cheng Kung University
Department of Biotechnology and Bioindustry Sciences
1, University Rd,
East Dist.,
Tainan city, Tainan 701
Taiwan

Dear Dr. Lin,

Thank you for submitting your Research Article entitled "Crystal structures of dimeric and heptameric mtHsp60 reveal the mechanism of chaperonin inactivation". It is a pleasure to let you know that your manuscript is now accepted for publication in Life Science Alliance. Congratulations on this interesting work.

DISTRIBUTION OF MATERIALS:

Again, congratulations on a very nice paper. I hope you found the review process to be constructive and are pleased with how the manuscript was handled editorially. We look forward to future exciting submissions from your lab.

Sincerely,
